# FILTERING WITH CONFIDENCE: WHEN DATA AUGMENTATION MEETS CONFORMAL PREDICTION

## ABSTRACT

With promising empirical performance across a wide range of applications, synthetic data augmentation appears a viable solution to data scarcity and the demands of increasingly data-intensive models. Its effectiveness lies in expanding the training set in a way that reduces estimator variance while introducing only minimal bias. Controlling this bias is therefore critical: effective data augmentation should generate diverse samples from the same underlying distribution as the training set, with minimal shifts. In this paper, we propose conformal data augmentation, a principled data filtering framework that leverages the power of conformal prediction to produce diverse synthetic data while filtering out poor-quality generations with provable risk control. Our method is simple to implement, requires no access to internal model logits, nor large-scale model retraining. We demonstrate the effectiveness of our approach across multiple tasks, including topic prediction, sentiment analysis, image classification, and fraud detection, showing consistent performance improvements of up to 40% in $F_1$ score over unaugmented baselines, and 4% over other filtered augmentation baselines.

## 1 INTRODUCTION

*Synthetic data augmentation* refers to a set of machine learning techniques and heuristics designed to artificially expand a training dataset Shorten & Khoshgoftaar (2019); Taqi et al. (2018). As noted by Huang et al. (2022), practitioners have long relied on augmenting inputs with perturbed versions of the original data—both to enhance model robustness to small perturbations and based on the general intuition that "more data is always better." With the emergence of advanced foundation models capable of generating remarkably high-quality synthetic data however (from images (Ho et al., 2020; Karras et al., 2017; 2019; Ramesh et al., 2022; Rombach et al., 2022), to text (Brown et al., 2020; Li et al., 2022; Touvron et al., 2023), or molecular structures (Jin et al., 2018; Shi et al., 2020)), synthetic data generation has experienced renewed interest. Such approaches promise significant practical advantages, particularly in reducing the time, cost, and effort involved in augmenting datasets through additional data collection and annotation Nadas et al. (2025). Synthetic data augmentation has already demonstrated promising empirical results across a wide range of applications. In natural language processing, it has been effectively used for model fine-tuning on small datasets and in low-resource language settings (Feng et al., 2020; Yang et al., 2019; Li et al., 2020; Mahamud et al., 2023; Wang et al., 2022), as well as for knowledge base construction (Li et al., 2024b) , etc. In computer vision, it has shown benefits in tasks such as image classification (He et al., 2016; Li et al., 2025) and object detection (Bochkovskiy et al., 2020), etc.

From a theoretical perspective, much is still to determine about the benefits of synthetic data. Recent theoretical insights from Huang et al. (2022); Nakada et al. (2024) have begun characterizing the effect of synthetic oversampling in certain regimes on estimator error bounds. Intuitively, synthetic oversampling should work well if it manages to enlarge the training set, reducing estimator variance whilst only incurring a slightly increased bias. Synthetic augmentation methods thus face a fundamental tension. On one hand, generated samples should closely follow the distribution of the original data to minimize bias—typically requiring using lower variability in the generation (or a "low temperature") to ensure that the generated data remains faithful to the original. On the other hand, synthetic samples need to be sufficiently diverse and decorrelated to be treated effectively as independent observations, a goal typically achieved by increasing generation variability (e.g., raising the temperature parameter) Havrilla et al. (2024).

Despite the current enthusiasm for synthetic data sampled from generative AI models, no principled approach has yet been proposed to determine this trade-off systematically Jordon et al. (2022). In fact, current methods for generating synthetic data exhibit limited flexibility in their handling of samples with varying levels of quality. To adapt the loss to various levels of synthetic data quality, some techniques, such as the approach by Jaine et al. Jain et al. (2024) or that of Nakada et al. Nakada et al. (2024), introduce hyperparameters to control the weights placed on the reconstruction errors corresponding to the original data and the synthetic data respectively, effectively putting less emphasis on the synthetic data if its quality is too low. But these methods are inherently inflexible and treat all generated datapoints similarly. In particular, these methods are unable to distinguish between good and bad synthetic examples, thereby effectively discarding all synthetic data points from distributions that produce mixtures of high-quality and low-quality outputs Alaa et al. (2022); Rajeswar et al. (2023); Ravuri & Vinyals (2019) . Finer methods, capable of operating effectively in high-variability ("high-temperature") regimes and explicitly distinguishing high-quality generated samples from poor ones, are still lacking.

**Contributions.** To bridge this gap, we introduce a principled filtering approach that selectively retains high-quality outputs with provable guarantees. Our method operates as a wrapper around existing generative AI-based data augmentation frameworks, enabling their use in high-temperature (high-variability) settings, while ensuring the quality of the generated content through conformal risk prediction. Specifically, our contributions include:

1. A principled framework (Section 2) for evaluating the quality of generated content, consisting of two primary components:
   (a) A scoring function that quantifies the quality of generated samples.
   (b) A rejection threshold that specifies the minimum acceptable quality, calibrated using conformal risk prediction (Section 3.1).
2. Provable guarantees of control of our procedure over the number of poor quality samples accepted in the augmented data using approximate conditional coverage in our setting (Section 3.2). Our method adapts the framework of Gibbs et al. (2025); Cherian et al. (2024) to provide robust, condition-specific quality guarantees.

Our approach is practical and straightforward to implement, requiring neither access to internal model logits nor extensive retraining. To evaluate the validity and practical utility of our method, we demonstrate its application across three text-based use cases and further assess its performance on three tabular datasets and one image dataset (Section 4). Across these tasks, our method consistently yields measurable improvements in downstream applications, including text classification, sentiment analysis, fraud detection, and image classification.

## 2 BACKGROUND: SYNTHETIC DATA GENERATION AND FILTERING

Let $h : \mathcal{X} \to \Omega$ denote a pretrained generative model (e.g., ChatGPT, Gemini or DALL·E, or any VAE-type of model fit to the data). Here $\mathcal{X}$ refers to a set of features on which to condition the generation, and $\Omega$ to the generation domain (e.g. space of images, documents, etc). While this paper mostly considers text and tabular data examples, our methodology can, in principle, extend to any domain where data can be generated using generative models. Consider a dataset $\mathcal{D} = \{X_i\}_{i=1}^{N}$, where each $X_i$ corresponds to a sample point (i.e. a document or image) and $N$ is the total number of samples. Our objective is to leverage $h$ to create alternative versions of each data point $X_i$, thereby increasing the dataset size. This approach is particularly useful in low-sample scenarios, such as when the training dataset is small (Section 4.1), as a mitigator of extreme class imbalance (Section 4.2).

**LLM-based Data Augmentation.** Ding et al. (2024) categorize LLM-based data augmentation into four classes: data creation, data reformation, data labeling, and human-LLM co-annotation. Our work specifically focuses on data reformation, where existing data points are transformed to produce new examples or enrich existing data points. Historically, reformation methods relied predominantly on rule-based approaches, such as token perturbations or back-translation. However, recent advancements in generative models have enabled significantly more diverse augmentation strategies. In this paper, we propose using a generative model based augmentation method due to its demonstrated ability to produce greater generative diversity. A detailed literature review of LLM-based augmentations is provided in Appendix B.

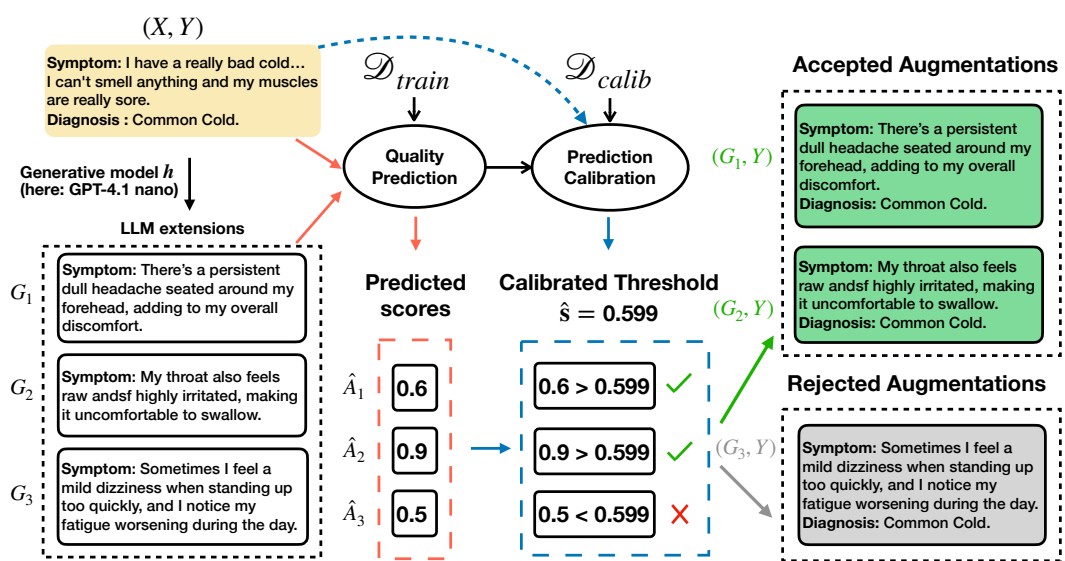

Figure 1: Illustration of the workflow in clinical disease prediction. Data augmentation candidate outputs from the generative model $h$ (GPT-4.1 nano in this example) are filtered by a quality predictor trained on $\mathcal{D}_{train}$ with a threshold calibrated by $\mathcal{D}_{\text{calib}}$. The retained output preserves the meaning of "common cold," while the discarded output does not correspond to the intended symptom.

Formally, let $X_i \in \mathcal{D}$ denote an observed data point, $Y_i \in \mathcal{Y}$ denote additional sample meta information (such as labels captions) which we might want to condition upon in our generating procedure. We assume that the data point $X_i$ is sampled from a true underlying distribution $h^\star$ that depends on the context/label: $X_i \sim h^\star(C_i, Y_i)$ where $C_i$ represents the latent context. Intuitively, $Y_i$ encodes observable attributes such as class labels or side information, and $C_i$ captures hidden structure or nuisance variation specific to the dataset at hand, and that is not directly observed but decides how $X_i$ is realized. To synthesize new instances from the same distribution, we generate $K$ alternative versions of $X_i$ by reusing $X_i$ as proxy for the latent context $C_i$:

$$(G_{ik})_{k=1}^K \sim h(X_i, Y_i, \tau),$$

where $h$ denotes the generative model conditioned explicitly on the observed data point and features $(X_i, Y_i)$, and $\tau$ is a temperature parameter controlling the model's generation variability. Thus, the generative model $h$ serves as an approximation to the true distribution $h^\star$, replacing the inaccessible latent context $C_i$ with observable surrogates $X_i, Y_i$.

**Evaluating Generation Quality**    Although effective, synthetic data from generative models can be noisy or distributionally shifted Feng et al. (2021); Kumar et al. (2020) — particularly when increasing the temperature $\tau$—, potentially reducing downstream performance. Various approaches, such as prompt engineering, direct generative modeling, retrieval-based methods, and filtering strategies (e.g., human evaluation, similarity metrics, classification-based filtering), have been proposed to improve synthetic data quality Alaa et al. (2022); Lewis et al. (2020); Liu et al. (2023). However, these filtering methods critically depend on accurate and oftentimes expensive quality metrics (such as human evaluation), which remain challenging Ding et al. (2024); Rajeswar et al. (2023). With increased generation diversity, ensuring quality becomes critical.

We propose revisiting here a simple filtration technique, as proposed in Islam et al. (2024); Kang et al. (2021); Li et al. (2024a). These methods all operate on the following premise: low-quality generations should be filtered out. Let $\mathcal{A} : \Omega \times \Omega \times \mathcal{Y} \to \mathbb{R}$ be a measure of a generated sample's quality. Ideally, $\mathcal{A}$ should quantify the degree of deviation of the generation from the underlying data distribution. Filtering-based methods choose to remove generated examples for which $\mathcal{A}(G_{ik}, X_i, Y_i) < \lambda$, for a user-defined threshold $\lambda$. The threshold $\lambda$ should be neither too low (to avoid content of low quality), nor too high (to avoid generating trivial rephrasings).

While this framework promises to improve the quality of data augmentation, it relies on access to a trustworthy evaluation metric $\mathcal{A}$. Choosing an unsuitable $\mathcal{A}$ can distort the training distribution. For example, simply measuring similarity between generated and original samples risks biasing the augmented data toward reproducing existing examples rather than capturing the broader distribution. Quality annotators might not necessarily exist, or if they do (e.g. human annotators in certain settings), they might be too expensive to deploy at scale. In the absence of gold-standard evaluations, the only option is to use a cheaper evaluator $\widehat{\mathcal{A}}$ (e.g. an LLM to evaluate text generations), thereby providing an imperfect, noisy surrogate for $\mathcal{A}$. Developing an approach that explicitly accounts for this noisiness and its uncertainty is therefore essential.

In this paper, we propose to adjust for the noisiness in the data by calibrating the acceptance threshold using conformal prediction. Rather than simply accepting the claim based on the quality metric $\widehat{\mathcal{A}}$, we propose calibrating the threshold $\lambda$ to mimic an oracle gold-standard $\mathcal{A}$ whilst limiting the number of false acceptances.

As a concrete example, Figure 1 illustrates our method's workflow in the context of clinical disease prediction. The input (a description of symptoms) is first processed by the generative model $h$ which is prompted to extend the description, after which the candidate outputs are screened using the quality evaluator $\widehat{\mathcal{A}}$ and a calibrated filtering threshold $\hat{s}$. The selected generations retain the intended meaning of "common cold," though minor surface errors such as typos may remain. Such typos can also be viewed as a form of data augmentation: while they slightly perturb the text, they preserve semantic meaning and can improve model robustness. By contrast, the discarded output fails to capture relevant symptoms of the common cold.

## 3 METHOD: FILTERING USING CONDITIONAL CONFORMAL RISK CONTROL

We propose a two-step approach for filtering outputs. In the first step, we randomly select a subset of the data, denoted by $\mathcal{D}_{\text{calib}} = \{(X_i, (G_{ik})_{k=1}^K, Y_i)\}$, on which evaluate the generations using both a gold-standard quality measure $\mathcal{A}$ and its surrogate $\widehat{\mathcal{A}}$ (for settings where no gold-standard exists, we propose an alternative in Section 3.3). This calibration set is then used to train a conformal prediction algorithm that calibrates the thresholding level $\lambda$ correctly for that particular generation, accounting for the uncertainty in $\widehat{\mathcal{A}}$ as a surrogate for $\mathcal{A}$. In the second step, we apply the conformal prediction filter—using the calibrated threshold—to the remaining dataset, $\mathcal{D}_{\text{aug}} = \{(X_i, (G_{ik})_{k=1}^K, Y_i)\}$, using the conformal prediction algorithm.

Let the sizes of $\mathcal{D}_{\text{calib}}$ and $\mathcal{D}_{\text{aug}}$ be $n_{\text{calib}}$ and $n_{\text{aug}}$, respectively. With a slight abuse of notation, we also use $\mathcal{D}_{\text{calib}}$ and $\mathcal{D}_{\text{aug}}$ to refer to the corresponding index sets when the meaning is clear from context.

### 3.1 CONTROLLING THE NUMBER OF WRONG INCLUSIONS

**Problem Formalization.** We consider the gold standard quality scores $\mathbf{A}_i = (A_{ik})_{k=1}^K$ and the corresponding surrogate scores $\hat{\mathbf{A}}_i = (\hat{A}_{ik})_{k=1}^K$ for the generations in the calibration data. We define the filtered set at surrogate level $s$ by the notation: $\mathcal{S}(\hat{\mathbf{A}}_i, s) = \{G_{ik} : \hat{A}_{ik} \geq s\}$. Let $\mathcal{L}_\lambda(\mathcal{S}(\hat{\mathbf{A}}_i, s), \mathbf{A}_i)$ denote a loss function that measures the quality of filtered output compared to the ground truth $\mathbf{A}_i$. For instance, we may define $\mathcal{L}_\lambda(\mathcal{S}(\hat{\mathbf{A}}_i, s), \mathbf{A}_i)$ to be the number of generations $\hat{A}_{ik}$ with surrogate score at least $s$ but whose gold-standard scores $A_{ik}$ are below the nominal quality threshold $\lambda$:

$$\mathcal{L}_\lambda(\mathcal{S}(\hat{\mathbf{A}}_i, s), \mathbf{A}_i) = |\{G_{ik} \in \mathcal{S}(\hat{\mathbf{A}}_i, s) : A_{ik} < \lambda\}|. \tag{1}$$

We then define the non-conformity score as

$$S_i = S(\hat{\mathbf{A}}_i, \mathbf{A}_i) = \inf\{s : \mathcal{L}(\mathcal{S}(\hat{\mathbf{A}}_i, s), \mathbf{A}_i) \leq \rho\}, \tag{2}$$

where $\rho$ is a hyperparameter that represents the tolerance on the loss, or the maximal number of "false discoveries" per sample that we are willing to allow. In other words, we define the non-conformity score $S(\hat{\mathbf{A}}_i, \mathbf{A}_i)$ as the minimal threshold $s$ such that the filtered set $\mathcal{S}(\hat{\mathbf{A}}_i, s)$ contains only all the generations for $X_i$ with surrogate score $\hat{\mathbf{A}}_{ik} > s$, and at most $\rho$ of these generations have gold-standard scores $\mathbf{A}_{ik} < \lambda$.

In this paper, we formulate the problem of filtering generations based on imperfect surrogate quality scores $\hat{\mathbf{A}}$ as a calibration problem: we need to select the surrogate filtering threshold $s$ in a data-driven manner so as to ensure that $\mathbb{P}(\mathcal{L}_\lambda(\mathcal{S}(\hat{\mathbf{A}}_{i_0}; s_{i_0}), \mathbf{A}_{i_0}) \leq \rho) \geq 1 - \alpha$ for all $i_0 \in \mathcal{D}_{\mathrm{aug}}$ with some user-specified confidence level $\alpha \in (0, 1)$. To this end, we propose leveraging conformal prediction (CP) Vovk et al. (2005); Angelopoulos et al. (2022) for risk control. Conformal methods provide finite-sample, distribution-free guarantees by calibrating predictions using a hold-out validation set (see Appendix B for a more in-depth review). In our setting, we use the distribution of the scores $S(\hat{\mathbf{A}}_i, \mathbf{A}_i)$ to correctly calibrate our rejection threshold to ensure retaining quality content. Letting $\hat{s}_{i_0}$ be the output of the conformal prediction algorithm for each $\mathbf{X}_{i_0}$ (see the explicit formula in equation 7 in Appendix), we will solely accept generated examples with $\hat{A}_{i_0 k} > \hat{s}_{i_0}$.

## 3.2 Conditional Conformal Risk Control

While this setup is intuitive, one could argue that, like $\lambda$, the surrogate threshold $s$ might just as well be chosen using data splitting – making the conformal prediction step appear unnecessary. However, our setting is more challenging: the difficulty of the filtering problem varies across samples, and fixed validation-based thresholds cannot adapt to this heterogeneity. To address this, we incorporate sample-specific information and apply conditional conformal prediction, allowing the filtering procedure to adapt to the hardness of each instance and thereby provide more reliable control.

While conformal prediction can act as a wrapper around any method, it is a well-established fact that it is impossible to get conditional results Barber et al. (2021). To address this, we adopt the relaxation proposed by Gibbs et al. (2025), which designs a prediction set that satisfies the guarantee over a specified function class $\mathcal{F}$:

$$\mathbb{E}\left[f(X_{i_0}) \cdot \left(\mathbf{1}\{\mathcal{L}_\lambda(\mathcal{S}(\hat{\mathbf{A}}_{i_0}; s_{i_0}), \mathbf{A}_{i_0}) \leq \rho)\} - (1 - \alpha)\right)\right] = 0, \text{ for all } f \in \mathcal{F}. \tag{3}$$

To handle conditional coverage without prior structural information, we take $\mathcal{F}$ to be a reproducing kernel Hilbert space (RKHS) with an added intercept, following Gibbs et al. (2025). Given a positive-definite kernel $W : \Omega \times \Omega \to \mathbb{R}$ (e.g., Gaussian/RBF kernel), define

$$\mathcal{F} = \{f_W(\cdot) + \beta : g_W \in \mathcal{F}_W, \beta \in \mathbb{R}\}, \tag{4}$$

where $\mathcal{F}_W$ is the RKHS function class associated with $W$. The intercept $\beta$ guarantees the marginal coverage, while the RKHS term $f_W$ enables flexible, smooth calibration of the conformity scores $\{S_i\}_{i \in \mathcal{D}_{calib}}$ against covariates. As the following lemma shows, the resulting cutoff $\hat{s}_{i_0}$ for each data $i_0 \in \mathcal{D}_{aug}$ satisfies the conditional guarantee over a localization region around $X_{i_0}$.

**Lemma 1** *Consider the function class $\mathcal{F}$ as defined in Equation 4, and assume $\mathcal{D}_{calib} \bigcup \mathcal{D}_{aug}$ are i.i.d. . Suppose $\mathcal{L}_\lambda(\cdot, \cdot)$ is monotone (i.e. for any sets $\mathcal{S}_{i_0}^1 \subseteq \mathcal{S}_{i_0}^2$, it must be the case that $\mathcal{L}_\lambda(\mathcal{S}_{i_0}^1, \mathbf{A}_{i_0}) \leq \mathcal{L}_\lambda(\mathcal{S}_{i_0}^2, \mathbf{A}_{i_0}))$ and $\mathcal{L}_\lambda(\emptyset, \cdot) = 0$. Assume $W(x, \cdot)$ defines a density with respect to each $x \in \Omega$, and sample $X'_{i_0} \mid X_{i_0} = x \sim W(x, \cdot)$. Then for all $f \in \mathcal{F}$, $i_0 \in \mathcal{D}_{aug}$,*

$$\mathbb{P}\left(\mathcal{L}_\lambda(\mathcal{S}(\hat{\mathbf{A}}_{i_0}; \hat{s}_{i_0}), \mathbf{A}_{i_0}) \leq \rho \mid X'_{i_0} = x'_{i_0}\right) = 1 - \alpha - \frac{\gamma \mathbb{E}[\hat{f}_W^{\hat{s}_{i_0}}(x'_{i_0})]}{\mathbb{E}[W(X_{i_0}, x'_{i_0})]},$$

*where $\gamma$ is the hyperparameter and $\hat{f}_W^{\hat{s}_{i_0}} \in \mathcal{F}_W$ is the fitted RKHS function defined in equation 5.*

Due to the infinite dimensionality of the RKHS class, the achieved coverage departs from the nominal level $1 - \alpha$ by a gap of $\frac{-\gamma \mathbb{E}[\hat{f}_W^{\hat{s}_{i_0}}(x'_{i_0})]}{\mathbb{E}[W(X_{i_0}, x'_{i_0})]}$. from the nominal level $1 - \alpha$. However, this coverage gap is estimable, and can be quantified using the procedure proposed in Gibbs et al. (2025). The proof of Lemma 1 is shown in Appendix C.2

In practice, we rely on the fast implementation of this approximate conditional CP algorithm as provided in Anonymous (2026), which provides a fast alternative to the original algorithm of Gibbs et al. (2025).

---

**Algorithm 1** Conformal Filtering

---

**Require:** Reference evaluation $A$; surrogate evaluation $\hat{A}$; calibration dataset $\mathcal{D}_{\text{calib}} = \{(X_i, (G_{ik})_{k=1}^K, Y_i)\}$ augmentation dataset $\mathcal{D}_{\text{aug}} = \{(X_i, (G_{ik})_{k=1}^K), Y_i)\}$; quality level $\lambda$; loss function $\mathcal{L}$; contamination allowance $\rho$
1: Compute the reference score: $A_{ik} = \mathcal{A}(G_{ik}, X_i, Y_i)$, $\forall i \in \mathcal{D}_{\text{calib}}$.
2: Compute the surrogate score: $\hat{A}_{ik} \leftarrow \hat{A}((G_{ik})_{k=1}^K, X_i, Y_i)$, $\forall i \in \mathcal{D}_{\text{calib}} \cup \mathcal{D}_{\text{aug}}$.
3: Compute the non-conformity score associated with loss $\mathcal{L}$ and $\rho$, denoted as $S(\hat{\mathbf{A}}_i, \mathbf{A}_i)$, according to Equation 2, $\forall i \in \mathcal{D}_{\text{calib}}$.
4: **for** $i \in D_{\text{aug}}$ **do**
5:     Fit conditional conformal prediction to find $\hat{s}_i$
6:     Select generations : $\mathcal{S}(\hat{\mathbf{A}}_i, \hat{s}_i) = \{G_{ik} : \hat{A}_{ik} \geq \hat{s}_i\}$
7: **end for**
8:
**Ensure:** The selected generations $\{\mathcal{S}_i : i \in \mathcal{D}_{\text{aug}}\}$.

---

### 3.3 LEARNING TO RECOGNIZE QUALITY OUTPUTS ON $\mathcal{D}_{\text{TRAIN}}$

In the previous discussion, we focused on the setting where the gold standard measure $\mathcal{A}$ is directly available on a small subset of the data. We now extend our approach to scenarios in which only a surrogate measure $\tilde{\mathcal{A}}$ can be observed. We shall assume that the surrogate measure satisfies

$$\mathcal{A}(G_{ik}, X_i, Y_i) = \mathbb{E}_{\tilde{X}_i \sim h^\star(C_i, Y_i)} \left[ \tilde{\mathcal{A}}(G_{ik}, \tilde{X}_i, Y_i) \right],$$

In other words, the gold-standard is the population average of the observed surrogate, and conversely, $\tilde{\mathcal{A}}$ can be viewed as a specific realization of a random variable, centered at $\mathcal{A}$. For instance, in text data, embedding-based similarity metrics such as cosine similarity computed from BERT embeddings are widely used to capture semantic coherence Devlin et al. (2018); Zhang et al. (2020). In image data, similarity measures based on CLIP scores Radford et al. (2021) are effective for capturing both semantic alignment and stylistic similarity. These metrics typically compare each generation directly against its original sample, which can be viewed as a realization from the underlying distribution $h^\star$.

We propose reducing the variability of the surrogate $\tilde{\mathcal{A}}$ by applying a regression-based strategy that leverages similar samples to approximate the underlying expectation. By smoothing over similar samples, this learned approximation is expected to more closely reflect the ground-truth measure $\mathcal{A}$.

Let $\tilde{A}_{ik} = \tilde{\mathcal{A}}(G_{ik}, X_i, Y_i)$ and $A_{ik} = \mathcal{A}(G_{ik}, X_i, Y_i)$. We model $A_{ik}$ as:
$$A_{ik} = \eta(G_k, C_i, Y_i) + \epsilon_{ik}$$
where $\epsilon_{ik}$ denotes some centered noise, and where $\eta(G_{ik}, C_i, Y_i) = \mathbb{E}[\tilde{A}_{ik} | G_{ik}, C_i, Y_i]$ is the population quantity we would like to estimate.

In this setting, we split the data into $\mathcal{D}_{\text{train}}, \mathcal{D}_{\text{calib}}$, and $\mathcal{D}_{\text{aug}}$. We then train a regression model $\hat{\mathcal{A}} : (G_{ik})_{k=1}^K, X_i, Y_i \mapsto \hat{A}_{ik}$ on $\mathcal{D}_{\text{train}}$ to predict $\tilde{A}_{ik}$. The model takes as input the generated samples $(G_{ik})_{k=1}^K$ together with the observed $(X_i, Y_i)$ and outputs a predicted score. For example, in text data, $\hat{\mathcal{A}}$ may incorporate features such as the semantic relevance between $G_{ik}$ and $(X_i, Y_i)$, as well as generation entropy, a metric that has been used to quantify uncertainty in generated outputs and to detect hallucinations. We then calibrate $\hat{\mathcal{A}}$ using $\tilde{\mathcal{A}}$ as an unbiased estimator of $\mathcal{A}$, as described in Section 3.1.

## 4 EXPERIMENTS

To highlight the efficacy of our method, we propose three case studies: (a) a data enrichment setting, (b) an imbalanced classification setting, and (c) a very low-data regime with generations of heterogeneous quality. Our examples span different data types, from text, to images, to tabular data.

### 4.1 PREDICTION WITH LLM-AUGMENTED TRAINING DATA

We study our data augmentation pipeline for clinical text classification, focusing on mapping symptom descriptions to medical diagnoses (Gretel AI, 2024). The dataset $(X_i, Y_i)$ consists of 853 training

samples and 212 test samples, where $X_i$ denotes a symptom description and $Y_i$ is one of 22 possible diagnoses. Each training example $(X_i, Y_i)$ is augmented using a generative language model (GPT-4.1 nano (OpenAI, 2023)), which extends the original symptom description $X_i$ with five additional sentences $(G_{ik})_{k=1}^5$. From these extensions, we generate new samples that inherit the original label, yielding a total of 4,265 synthetic observations $\{(G_{ik})_{k=1}^5 : i = 1, 2, \cdots, 853\}$.

To ensure output quality, we employ a two-stage evaluation strategy. First, a random subset of 500 generations, derived from 100 symptom descriptions, is evaluated with a high-accuracy model $\mathcal{A}$ (Gemini-2.5-pro (Comanici et al., 2025)), forming the calibration set $\mathcal{D}_{\text{calib}}$. Next, all augmented samples are scored using a faster, lower-cost surrogate model $\hat{\mathcal{A}}$ (Gemini-2.5-flash (Comanici et al., 2025)). Both models assign a score in $[0, 1]$, with $0.5$ as the retention threshold (see the detailed prompt in the supplement). This design reflects a practical labeling scenario in which reliable annotations are costly, whereas approximate labels can be obtained inexpensively. Let $(A_{ik})_{i \in \mathcal{D}_{\text{calib}}}$ denote the Gemini-pro scores and $(\hat{A}_{ik})_{i=1}^{853}$ denote the Gemini-flash scores.

We then apply our calibration step. For each $(X_i, (G_{ik})_{k=1}^K, Y_i) \in \mathcal{D}_{\text{calib}}$, a non-conformity score is defined as the minimum threshold that guarantees all selected sentences achieve a pro-score above $0.5$, so that: $S(\hat{\mathbf{A}}_i, \mathbf{A}_i) = \inf \left\{ \tau : \left| \{ G_{ik} : 1 \le k \le 5, \hat{A}_{ik} \ge \tau, A_{ik} < 0.5 \} \right| \le 1 \right\}$.

For each $X_i$, we embed the text into a lower-dimensional space using Latent Dirichlet Allocation (LDA) (Blei et al., 2003), a classical method for producing low-dimensional text representations, fitted on the entire training set. Let $\hat{\pi}(\cdot)$ denote the resulting LDA mapping. We construct a kernel $W(\cdot, \cdot) = \exp\{-\xi \|\hat{\pi}(\cdot) - \hat{\pi}(\cdot)\|_2^2\}$, with $\xi$ selected via cross-validation. Then we apply conditional CP (CondCP) (Gibbs et al., 2025) with $\alpha = 0.1, \rho = 0$ to obtain adaptive thresholds on $\hat{A}_{ik}$.

To evaluate performance, we fine-tune a diagnostic classifier (distilbert-base-uncased (Devlin et al., 2018)) using LoRA (Hu et al., 2022). Each training iteration consists of 100 fixed high-confidence documents (selected by the pro-scores) and 400 additional documents sampled under one of the following filtering schemes: (1) No augmentation; (2) No filtering; (3) Filtering by $\hat{A}_{ik}$ only (threshold = 0.5); (4) Hybrid filtering (using $A_{ik}$ for $\mathcal{D}_{\text{calib}}$ and $\hat{A}_{ik}$ with threshold 0.5 for the remainder); (5) CondCP-based filtering on $\mathcal{D}_{\text{aug}}$ (using $A_{ik}$ for $\mathcal{D}_{\text{calib}}$).

Performance, averaged across 20 trials, is reported in Figure 2, with evaluation consistently conducted on the held-out test set. We also report the results of experiments performed in an identical manner on topic prediction (predicting the topic of statistical abstracts downloaded from arXiv with 5 possible categories) and sentiment analysis (predicting one of 6 emotions on a dataset of Twitter messages — see details in the Appendix D). Overall, across these three datasets, our CondCP filter improves the precision, recall, and F1-score by up to 3% over the unaugmented baseline, and substantially improves upon the unfiltered baseline. We note that in the diagnosis task, the unfiltered augmentation outperforms the unaugmented baseline, but this advantage does not hold for the abstract and Twitter datasets, suggesting that including all generations can be detrimental when low-quality samples are present. In contrast, the CondCP filter achieves the best performance across all metrics and tasks.

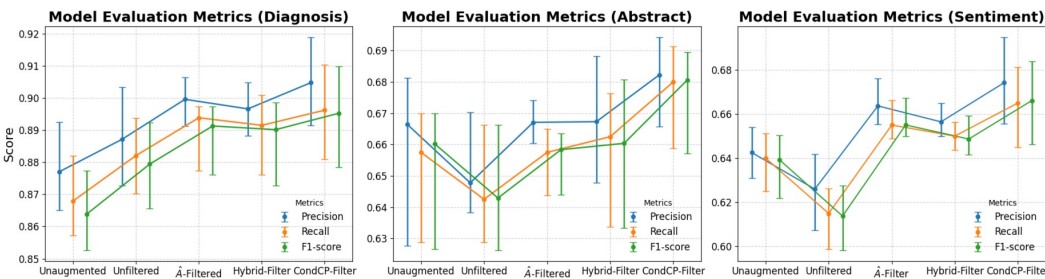

Figure 2: Evaluation of different data augmentation methods on diagnosis prediction, abstract topic prediction, and Twitter message sentiment prediction. Results are averaged over 20 replicates. Error bars indicate the interquartile range, with centers representing the median and boundaries corresponding to the first and third quartiles.

| Dataset | Strategy | $F_1$ (↑) | Precision (↑) | Recall (↑) | Stable Rank (↑) |
|---|---|---|---|---|---|
| Thyroid (N=2,644, d=27, Imb.=6.4%) | Unaugmented | $0.139 \pm 0.080$ | $\mathbf{0.538 \pm 0.240}$ | $0.081 \pm 0.050$ | $7.713 \pm 0.169$ |
| | SMOTE | $0.499 \pm 0.022$ | $0.354 \pm 0.017$ | $\mathbf{0.848 \pm 0.061}$ | $7.358 \pm 0.202$ |
| | Unfiltered | $0.495 \mp 0.031$ | $0.356 \pm 0.030$ | $\underline{0.819 \pm 0.061}$ | $8.238 \pm 0.812$ |
| | $\widehat{A}$-Filter | $\underline{0.507 \pm 0.046}$ | $0.370 \pm 0.046$ | $0.817 \pm 0.065$ | $\underline{8.495 \pm 0.384}$ |
| | CondCP-Filter | $\mathbf{0.542 \pm 0.043}$ | $\underline{0.417 \pm 0.043}$ | $0.783 \pm 0.070$ | $\mathbf{8.730 \pm 0.336}$ |
| Credit Card Fraud (N=284,807, d=28, Imb.=0.17%) | Unaugmented | $\underline{0.732 \pm 0.023}$ | $\mathbf{0.886 \pm 0.044}$ | $0.626 \pm 0.045$ | $\mathbf{25.790 \pm 0.605}$ |
| | SMOTE | $0.108 \pm 0.004$ | $0.057 \pm 0.002$ | $\mathbf{0.920 \pm 0.023}$ | $2.405 \pm 0.012$ |
| | Unfiltered | $0.709 \pm 0.029$ | $0.668 \pm 0.049$ | $0.760 \pm 0.055$ | $1.962 \pm .049$ |
| | $\widehat{A}$-Filter | $0.711 \pm 0.030$ | $0.675 \pm 0.048$ | $0.757 \pm 0.061$ | $2.273 \pm 0.117$ |
| | CondCP-Filter | $\mathbf{0.807 \pm .027}$ | $\underline{0.813 \pm 0.041}$ | $\underline{0.803 \pm 0.045}$ | $\underline{7.380 \pm 1.049}$ |
| MNIST 7 vs. Others (N=70,000, d=784, Imb.=10.9%) | Unaugmented | $0.894 \pm 0.010$ | $\mathbf{0.905 \pm 0.008}$ | $0.882 \pm 0.027$ | $\mathbf{16.582 \pm 0.065}$ |
| | SMOTE | $0.880 \pm 0.008$ | $0.858 \pm 0.001$ | $\mathbf{0.903 \pm 0.015}$ | $13.507 \pm 0.085$ |
| | Unfiltered | $0.891 \pm 0.011$ | $0.891 \pm 0.005$ | $\underline{0.892 \pm 0.027}$ | $\underline{11.980 \pm 0.840}$ |
| | $\widehat{A}$-Filter | $0.892 \pm 0.009$ | $0.895 \pm 0.008$ | $0.888 \pm 0.025$ | $11.865 \pm 0.855$ |
| | CondCP-Filter | $\mathbf{0.896 \pm 0.007}$ | $\underline{0.904 \pm 0.005}$ | $0.888 \pm 0.025$ | $11.972 \pm 1.510$ |

Table 1: Results of imbalanced classification: predictive performance metrics and data diversity (Stable Rank) averaged over 10 different splits. Dataset sizes (N), feature dimensions (d), and imbalance rates (Imb.) are given in parentheses. Higher values are better for all metrics presented. The best value is bolded and the second best value is underlined. Across all benchmarks, our CondCP-Filter consistently attains the best $F_1$ and increase data diversity, as reflected by higher stable rank.

### 4.2 IMBALANCED CLASSIFICATION: TABULAR DATA EXAMPLES

In imbalanced classification, models often default to predicting the majority class, yielding misleadingly high accuracy while missing rare but critical events. For example, in the European Credit-Card Fraud dataset[1] (0.17% frauds), labeling all cases as "non-fraud" achieves 99.8% accuracy but detects no fraud (He & Garcia, 2009; Japkowicz & Stephen, 2002). Data augmentation seems therefore a promising way of enhancing recall whilst maintaining precision.

We evaluate our method on three benchmark datasets spanning different imbalance regimes: European Credit-Card Fraud (Kaggle), Thyroid (OpenML), and MNIST-7 vs Others (OpenML). See the details of the dataset and experiment setup in Appendix D.8.1. In these settings, to generate new data, we train a Variational AutoEncoder (VAE) (Kingma & Welling, 2013; Sohn et al., 2015) to increase the number of samples from the minority class. Since gold-standard quality measures are not available in this setting, we use the procedure detailed in section 3.3, and use for our surrogate scores $\hat{A}$ a gradient boosting predictor, trained to predict the surrogate measure $\tilde{A}$. For the experiments presented in this subsection, $\tilde{A}$ is defined as the geometric mean of a $k$-nearest-neighbor similarity (to measure closeness to real minority data) and a cosine similarity (directional closeness to the reference data).

In each case, we split the data into train/calibration/test subsets (60/20/20) and report average $F_1$ scores. We fit a logistic regression classifier, and we compare the performance of our CP-filtering procedure with (a) an unaugmented baseline; (b) SMOTE (Chawla et al., 2002), a widely used oversampling method that interpolates minority examples in feature space; (c) unfiltered augmentation; and (d) various filtering procedures (e.g. CP-based filtering, and filtering based on $\hat{A}$).

Table 1 reports $F_1$, precision, recall, and Stable Rank across five benchmarks. On *severely imbalanced datasets* such as credit-card fraud, quality-controlled augmentation clearly dominates both the unaugmented baseline and SMOTE; while SMOTE boosts recall, it inflates false positives, lowering precision. Our $\widehat{A}$-Filter and CondCP-Filter maintain recall while improving precision, yielding the best $F_1$. On *moderate imbalance* (Thyroid), all methods perform similarly, but our filters still outperform baselines and increase Stable Rank, indicating genuine diversity rather than duplication. For MNIST-7, where *the signal is strong*, unfiltered augmentation already works well; nonetheless, CondCP-Filter achieves the highest $F_1$ and superior precision–recall balance, showing the benefit of targeted acceptance even in easier tasks.

Beyond predictive performance, we also study diversity metrics of the training sets after augmentation with filtering. In particular, we compute the *stable rank* of the feature matrix $(X)$, which is defined as $\|X\|_F^2/\|X\|_2^2$. Stable rank captures the effective dimensionality of the sample cloud (Tsitsulin et al., 2023). Whereas simple augmentation often inflates data density along a few dominant directions

---

[1]https://www.kaggle.com/datasets/mlg-ulb/creditcardfraud/data

(due to interpolation), our method introduces genuinely new modes in the minority manifold, reflected in higher stable rank. These results indicate that quality-controlled generation is not only effective for balancing datasets, but also enhances geometric richness in ways that may improve generalization.

### 4.3 LOW-DATA REGIME WITH MIXED-QUALITY GENERATIONS: AN IMAGE ANALYSIS EXAMPLE

In this example, we wish to evaluate the performance of our method in a low-data regime for classification, with inputs of mixed qualities (with a distribution of 50% good inputs, 50% bad). In this setting, we expect the filtering procedure to be particularly useful.

To this end, we consider two classes from the mini ImageNet dataset (arctic foxes and toucans), and make a dataset of around 172 training images (86 per class). A moderate-capacity CNN is trained from scratch on this base set. We simulate data augmentation by masking 70% of each training image and asking DALL·E 2 to inpaint the missing regions, and replacing the masked area by the generated content (Fig. 3). To simulate unhelpful data generations, we use the masks as additional generations that need filtering out.

Each candidate is assigned a surrogate quality score. To compute this score, we first train the CNN on a separate split of the data (with around 35 images per class on average), and take the CNN's class–probability margin $|p(y \mid x) - 0.5|$ as a measure of the compatibility of the generation with its class and the image clip score as the quality gold standard. We then compare three filtering regimes: (a) **Threshold baseline:** keep candidates with score $\geq \lambda$; (b) **Marginal CP:** compute a global cutoff from calibration documents using split–conformal quantiles of per–document scores $S_{\text{doc}}$; and (c) **Conditional CP:** compute adaptive per–document cutoffs from PCA embeddings of the base images using the CondCP-filter procedure.

After selection, we retrain a CNN from scratch on all original images plus the selected augmentations. Validation and test sets remain fixed. The validation set (175 images of each class) is used for hyperparameter selection (here, the threshold $\lambda$, chosen in the grid $\lambda \in \{0.5, 0.75, 0.8, 0.9\}$). The CP parameters are fixed to $\rho = 0$ and $\alpha = 0.1$). Results are reported on the test set (300 images), and averaged over 5 runs of the procedure, shuffling the training set split into calibration and testing. Across this two–class task, conformal filtering yields consistent improvements, with the CondCP filter providing a $+3.7\%$ accuracy improvement over the unaugmented baseline and $+2\%$ gain in test accuracy over the baseline. Importantly, we note that the marginal CP baseline does not yield any improvement in accuracy, highlighting the importance of using conditional CP in this setting. Moreover, we do note the importance of filtering just the right amount, as the choice of the $\lambda$ does not default to the minimal value.

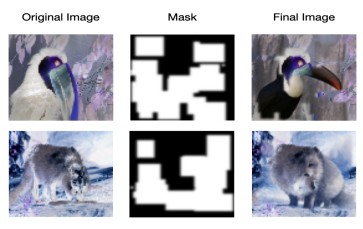

| | Original Image | Mask | Final Image |

Figure 3: Examples of data generation procedures for an image of a Toucan (top row) and an image of an Arctic Fox.

| $\lambda$ | Regime | Training Set Size | Test Accuracy |
|---|---|---|---|
| - | Unaugmented | $172.8 \pm 5.8$ | $0.786 \pm 0.023$ |
| - | Unfiltered | $3630.8 \pm 317.4$ | $0.802 \pm 0.006$ |
| 0.80 | $\hat{A}$ Filter | $897.0 \pm 986.9$ | $0.804 \pm 0.036$ |
| 0.75 | CondCP Filter | $1916.6 \pm 158.4$ | $\mathbf{0.823 \pm 0.018}$ |
| 0.50 | MargCP Filter | $1334.8 \pm 463.2$ | $0.763 \pm 0.055$ |

Figure 4: Performance of the methods on the ImageNet dataset, averaged over 5 iterations.

## 5 CONCLUSION

In this study, we propose a principled data augmentation algorithm that evaluates the quality of generated content beyond simple comparison with observed data, and filters out low-quality generations with provable risk control. Future directions for improvement include: (1) extending our methodology to other generative settings such as counterfactual or retrieval-based augmentation; (2) integrating our framework with other conformal prediction techniques, such as adaptive level control for different tasks.

## ETHICS STATEMENT

All authors confirm that this work adheres to the ICLR Code of Ethics. This research does not involve human subjects, personally identifiable information, or sensitive data. The datasets used are publicly available and have been properly cited. No potentially harmful applications, discriminatory outcomes, or security/privacy risks are anticipated from this study.

We have made efforts to ensure fairness, transparency, and integrity in both the methodology and the interpretation of results. All analyses and conclusions comply with ethical standards for reproducibility, research documentation, and integrity. No conflicts of interest or sponsorship influenced this work.

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

## A  THE USE OF LARGE LANGUAGE MODELS (LLMS)

In this work, LLMs were used for synthetic data generation as part of our research on data augmentation with generative models. Specifically, LLMs produced candidate text samples that were subsequently filtered, evaluated, and integrated into the experimental pipeline. The role of LLMs was limited to data generation within the proposed methodology and did not extend to research ideation, conceptual framing, or substantive writing of the manuscript.

All analysis, interpretation, and writing were conducted by the authors. We take full responsibility for the content of this paper, including any outputs derived from LLMs. No portion of the manuscript relies on fabricated or plagiarized material produced by LLMs.

## B  RELATED LITERATURES

**Data Augmentation**  In Ding et al. (2024), LLM-based data augmentations are categorized into four categories: data creation, data reformation, data labeling, and human-LLM co-annotation. In this work we focus on the data reformation, which transforms existing data to produce new data. People have proposed data reformation approaches prior to the advent of pre-trained generative models, with the majority of them being rule-based methods (Feng et al. (2021)). For instance, Easy Data Augmentation (EDA) Wei & Zou (2019) applies token-level perturbations like synonym replacement, random insertion, deletion, and swapping; Machine back-translation involves translating the original sentences into another language and then translating them back to the original language Sennrich et al. (2015); Edunov et al. (2018). Model-based methods, by contrast, leverage generative models to synthesize new text. Common examples include paraphrasing Kumar et al. (2019), semantic text exchange Feng et al. (2019) and masked word prediction followed by replacement Ng et al. (2020). The goal is to generate synthetic data that introduces diversity while maintaining semantic consistency (often referred to as label-preserving in classification problems (Xie et al., 2020)). Ideally, augmented data should not be too similar to the original (which limits diversity) nor too dissimilar (which risks domain shift and degraded performance).

Despite its ease of implementation, synthetic data generated by generative models is often noisy and distributionally misaligned with the original data, potentially hindering model training (Zhang et al., 2022). To address this, several complementary strategies have been proposed. Some approaches focus on prompt engineering to steer generation more precisely (Veselovsky et al., 2023; Gupta et al., 2023), while others leverage model-based augmentation by estimating a generative process from the training set and sampling from it (Anaby-Tavor et al., 2020). Retrieval-based techniques further enhance the expressiveness of LLM-driven augmentation by incorporating external knowledge (Chai et al., 2025). At generation time, diffusion-based models have been guided toward low-density or underrepresented modes—such as through class-conditional or classifier-informed sampling for minority classes (Sehwag et al., 2022; Trabucco et al., 2023). Prompt perturbation has been used to mitigate semantic ambiguity and encourage coverage of diverse outputs (Sarıyıldız et al., 2023; Shipard et al., 2023). In parallel, foundation models have been fine-tuned to better align with the target domain, either via on-the-fly adaptation (e.g., GenDataAgent (Li et al., 2025)) or through large-scale domain-specific retraining, as demonstrated with text-to-image diffusion models on ImageNet-scale data (Azizi et al., 2023; Dunlap et al., 2023).

These approaches improve the quality of the generations, but still, there could be low-quality generations that ideally we would like to filter out. The filtering-based methods evaluate typically the generations based on quality metrics, such as human evaluation (Wang et al., 2021; Liu et al., 2022)(which can be expensive), model confidence or difficulty (Hemmat et al., 2023; Agarwal et al., 2022), similarity to the original input in paraphrasing Li et al. (2024b), confidence of LLM, or classifiers trained to distinguish real from synthetic data Veselovsky et al. (2023). Most of these methods either explicitly or implicitly leverage the prediction on the quality of the generations, which could be problematic when the prediction is not accurate.

**Filtering.**  Filtering methods are commonly based on the following strategies (Chai et al., 2025):

- **Lexical overlap:** filtering based on n-gram overlap metrics such as ROUGE.
- **Semantic similarity:** filtering based on cosine similarity in embedding space.

- **Model-based filtering:** scoring generations using pre-trained models (e.g., LLMs).

- **Round-trip consistency:** checking whether back-translation or round-trip generation recovers the original input.

- **Influence-function filtering:** discarding augmentations predicted to harm downstream performance (Yang et al., 2020).

In practice, many augmentation pipelines combine multiple filters; for example, a heuristic may first remove obviously poor outputs, and then the top-$k$ most similar examples to the ground truth are retained (Chai et al., 2025). The goal is to balance fidelity and diversity: overly strict filters yield safe but low-diversity augmentations, while overly permissive filters risk introducing label noise or factual errors. Recent methods explicitly address this trade-off. For example, Mask-then-Fill (Gao et al., 2023) reports that infilling achieves a balance between novelty and distributional similarity to the source, likely through careful tuning of mask size and model parameters. In contrast, M4DA (Yao et al., 2024) promotes diversity by masking tokens to increase variance and then selecting variants with the highest semantic complexity. While the generated text must still preserve the original meaning, this preference for more complex rephrasings can yield stronger augmentation effects. Experiments on text classification benchmarks show that such methods can outperform conservative approaches, suggesting that filtering should not always default to the safest outputs—some controlled complexity, when consistent, is beneficial.

## C    RELATED WORKS ON CONFORMAL PREDICTION

**Conformal Prediction**    Given a dataset $\{(X_i, Y_i)\}_{i=1}^{N}$, a pretrained-predictor $h$ and a new text input $X_{n+1}$, conformal prediction (Vovk et al., 2005) attempts to construct a prediction $\hat{C}(X_{n+1})$ such that $\mathbb{P}\left(Y_{n+1} \notin \hat{C}(X_{n+1})\right) \leq 1 - \alpha$ for some user-specified $\alpha$. Conformal prediction has the distribution-free property and it is finite-sample valid under the exchangeability of the data points $\{(X_i, Y_i)\}_{i=1}^{n+1}$. For instance, in split conformal prediction, one can define $S_i = \|Y_i - h(X_i)\|$ and then set $\hat{C}(X_{n+1}) = \{y : \|y - h(X_{n+1})\| \leq \tau\}$ where $\tau = \text{quantile}(\{S_i\}_{i=1}^{n} \cup \{\infty\}, 1 - \alpha)$ . This type of method provides a guarantee on marginal coverage. Previous studies have demonstrated that achieving exact conditional coverage is impossible without any further distributionally assumption (Barber et al., 2021). Nevertheless, researchers have developed methods to achieve conditional coverage with controllable error rates (Zhang & Candès, 2024; Gibbs et al., 2025).

**Conformal Prediction and LLMs**    Researchers have increasingly explored the application of conformal prediction (CP) frameworks in generative models for factuality control, motivated by CP's ability to provide distribution-free inference. In (Ren et al., 2023; Kumar et al., 2023), CP is employed to identify probability thresholds for next-token generation, thereby selecting response candidates. Several works have proposed CP methods that do not require access to model logits (Su et al., 2024). For instance, (Shahrokhi et al., 2025; Quach et al., 2023) use CP to determine the number of generations needed to construct a prediction set that includes at least one truthful response or satisfies a specified confidence level. Other approaches, such as (Mohri & Hashimoto, 2024; Cherian et al., 2024), segment LLM outputs into individual claims and apply CP to select factual ones. Additionally, Gui et al. (2024) extends CP to multiple test units with a focus on ensuring valid false discovery rate (FDR) control. Despite the successes in these applications, how CP can be applied in data augmentation is under-explored, perhaps due to its unsupervised nature.

**Conditional Conformal Prediction**    While Conformal prediction seems like a promising wrapper around any blackbox method, its scope is fundamentally restricted to marginal coverage guarantees. However, marginal coverage does not preclude large variability in *conditional coverage*, defined as

$$\mathbb{P}\big(Y_{n+1} \in \hat{C}(X_{n+1}) \mid X_{n+1} = x\big) = 1 - \alpha,$$

which may differ significantly across inputs. This limitation is critical in sensitive applications (e.g., medicine, finance), where systematic under-coverage on certain subgroups undermines reliability. Prior work shows that in distribution-free settings, exact conditional coverage is impossible: any set satisfying it must degenerate to $\hat{C}(X_{n+1}) = \mathbb{R}$ with infinite expected size (Barber et al., 2021).

To address this, Gibbs et al. (2025) reformulate conditional coverage as a marginal constraint over measurable functions $f$:

$$\mathbb{E}\Big[f(X_{n+1}) \cdot \big(\mathbf{1}\{Y_{n+1} \in \hat{C}(X_{n+1})\} - (1-\alpha)\big)\Big] = 0.$$

They then restrict $f$ to a user-specified function class $\mathcal{F}$, yielding approximate conditional validity. Different choices of $\mathcal{F}$ lead to different notions of conditional coverage: for example, $\mathcal{F} = \{\text{constants}\}$ recovers marginal coverage, while $\mathcal{F} = \big\{\sum_{G \in \mathcal{G}} \beta_G \mathbf{1}\{x \in G\} : \beta \in \mathbb{R}^{|\mathcal{G}|}\big\}$ enforces group-conditional guarantees. Gibbs et al. (2025), by contrast, allow $\mathcal{F}$ to take more general forms, from linear distribution shifts, to more complex shifts parametrized by an RKHS function.

### C.1 Additional details on conditional conformal

In our setting, the conformity score $S_{i_0}$ is unknown for every $i_0 \in \mathcal{D}_{\text{aug}}$; we therefore impute a value $S$ for each such test index and solve a single regularized quantile problem that treats the imputed test pair symmetrically with the calibration data. Following Gibbs et al. (2025), we estimate a high–probability upper bound for these scores $\{S_i\}_{i \in \mathcal{D}_{\text{calib}}} \cup S$ by fitting a regularized kernel quantile regression:

$$\hat{f}_S = \arg\min_{f \in \mathcal{F}^*} \left\{ \frac{1}{|\mathcal{D}_{\text{calib}}|+1} \sum_{i \in \mathcal{D}_{\text{calib}}} \ell_\alpha(S_i - f(X_i)) + \frac{1}{|\mathcal{D}_{\text{calib}}|+1} \ell_\alpha(S - f(X_{i_0})) + \frac{\gamma}{2}\|f_W\|_W^2 \right\}, \tag{5}$$

where $\alpha \in (0,1)$, $\ell_\alpha(z) = (1-\alpha)[z]_+ + \alpha[z]_-$ is the pinball loss, $\gamma > 0$ is a regularization parameter, and $\|\cdot\|_W$ is the RKHS norm associated with the positive–definite kernel $W$.

By the representer theorem (Kimeldorf & Wahba, 1971), the optimizer admits the finite expansion

$$\hat{f}_S(X) = \hat{\beta}_S + \frac{1}{\gamma} \sum_{i \in \mathcal{D}_{\text{calib}} \cup \{i_0\}} \hat{v}_{S,i}\, W(X, X_i), \tag{6}$$

with coefficient vector $\hat{v}_S \in \mathbb{R}^{|\mathcal{D}_{\text{calib}}|+1}$ and intercept $\hat{\beta}_S \in \mathbb{R}$. Accordingly, the fitted RKHS component is of form $\hat{f}_W(x) = \frac{1}{\gamma} \sum_{i \in \mathcal{D}_{\text{calib}} \cup \{i_0\}} \hat{v}_{S,i}\, W(x, X_i)$ As shown in the discussion of Anonymous (2026), the coefficients $\hat{v}_S$ depend *affinely* on the imputed value $S$ and the mapping $S \mapsto \hat{v}_S$ is nondecreasing. Consequently, the event $S \le \hat{f}_S(X_{i_0})$ is equivalent to the linear inequality $\hat{v}_{S,i_0} \le 1 - \alpha$. Following the standard randomized conformalization in Anonymous (2026); Gibbs et al. (2025), we replace $1 - \alpha$ by a draw $U \sim \text{Unif}(-\alpha,\, 1-\alpha)$, and define the final fitted cutoff by

$$\hat{s}_{i_0} = \max\{ S : \hat{v}_{S,i_0} \le U \}. \tag{7}$$

Equivalently, the final prediction set $\hat{C}(X_{i_0})$ is obtained by plugging the cutoff $s = \hat{s}_{i_0}$ into the set construction $\mathcal{S}(\widehat{\mathbf{A}}_{i_0}; s)$.

**Coverage guarantee.** The following lemma collects the conditional guarantee delivered by this construction. For each $i_0 \in \mathcal{D}_{aug}$, we write $\hat{f}_W^{\hat{s}_{i_0}}$ for the fitted RKHS function evaluated at the cutoff $\hat{s}_{i_0}$.

**Lemma 2 (Coverage; cf. Gibbs et al. (2025); Cherian et al. (2024))** *Let $\mathcal{F}$ be as in equation 4, and assume the pooled indices $\mathcal{D}_{\text{calib}} \cup \mathcal{D}_{\text{aug}}$ are exchangeable. Suppose the loss $\mathcal{L}(\cdot, \cdot)$ is monotone in its first argument (i.e., if $\mathcal{S}_{i_0}^1 \subseteq \mathcal{S}_{i_0}^2$ then $\mathcal{L}(\mathcal{S}_{i_0}^1, \mathbf{A}_{i_0}) \le \mathcal{L}(\mathcal{S}_{i_0}^2, \mathbf{A}_{i_0}))$ and satisfies $\mathcal{L}(\emptyset, \cdot) = 0$. Then, for all $f \in \mathcal{F}$ and all $i_0 \in \mathcal{D}_{\text{aug}}$,*

$$\mathbb{E}\Big[f(X_{i_0})\Big\{\mathbf{1}\Big(\mathcal{L}_\lambda\big(\mathcal{S}(\widehat{\mathbf{A}}_{i_0}; \hat{s}_{i_0}), \mathbf{A}_{i_0}\big) \le \rho\Big) - (1-\alpha)\Big\}\Big] = -\gamma\,\mathbb{E}\Big[\langle \hat{f}_W^{\hat{s}_{i_0}},\, f_W\rangle_W\Big].$$

The lemma shows that the deviation from the nominal level $(1-\alpha)$ comes from the RKHS inner product involving the learned calibration function, yielding an estimable coverage gap as discussed in Gibbs et al. (2025).

## C.2 PROOF OF LEMMA 1

For the localized conformal prediction, we adapt Lemma 2 to a class of covariate shifts induced by the density kernel $W$. Under the setting of Lemma 1, the tuples

$$(X_1, \{G_{1k}\}_{k\in[K]}, Y_1), \ldots, (X_n, \{G_{nk}\}_{k\in[K]}, Y_n)$$

are independent of $(X_{i_0}, \{G_{i_0k}\}_{k\in[K]}, Y_{i_0}, X'_{i_0})$. By definition of $X'_{i_0}$, the joint distribution of $(X_{i_0}, \{G_{i_0k}\}_{k\in[K]}, Y_{i_0}, X'_{i_0})$ is given by

$$X_{i_0} \sim P_X, \qquad Y_{i_0} \mid X_{i_0} \sim P_{Y|X}, \qquad (G_{i_0k})_{k=1}^K \mid (X_{i_0}, Y_{i_0}) \sim h(X_{i_0}, Y_{i_0}, \tau),$$

$$X'_{i_0} \mid (X_{i_0}, (G_{i_0k})_{k=1}^K, Y_{i_0}) \sim W(X_{i_0}, \cdot),$$

so that $X'_{i_0} \perp\!\!\!\perp \big((G_{i_0k})_{k=1}^K, Y_{i_0}\big) \mid X_{i_0}$.

For any realization $x' \in \Omega$, Bayes' rule yields

$$\big(X_{i_0}, (G_{i_0k})_{k=1}^K, Y_{i_0}\big) \mid X'_{i_0} = x' \ \sim \ \frac{W(x, x')}{\mathbb{E}[\,W(X, x')\,]} \ dP_{(X,\mathbf{G},Y)}(x, \mathbf{G}, y),$$

i.e., the original joint distribution $P_{(X,\mathbf{G},Y)}$ tilted by the weight $W(x, x')$ and renormalized by $\mathbb{E}[W(X, x')]$.

Conditioning on $X'_{i_0} = x'$ and writing $\mathcal{S}(\widehat{\mathbf{A}}_{i_0}; s_{i_0})$ for the set construction, we obtain

$$\mathbb{E}\Big[\mathbf{1}\Big\{\mathcal{L}_\lambda\big(\mathcal{S}(\widehat{\mathbf{A}}_{i_0}; s_{i_0}), \mathbf{A}_{i_0}\big) \leq \rho\Big\} - (1-\alpha) \,\Big|\, X'_{i_0} = x'\Big]$$

$$= \frac{\mathbb{E}\Big[W(X, x') \cdot \Big(\mathbf{1}\Big\{\mathcal{L}_\lambda\big(\mathcal{S}(\widehat{\mathbf{A}}_{i_0}; s_{i_0}), \mathbf{A}_{i_0}\big) \leq \rho\Big\} - (1-\alpha)\Big)\Big]}{\mathbb{E}[\,W(X, x')\,]}$$

$$= \frac{-\gamma\, \mathbb{E}\Big[\big\langle \hat{f}_W^{\hat{s}_{i_0}}, W(\cdot, x')\big\rangle_W\Big]}{\mathbb{E}[\,W(X, x')\,]} \quad \text{(by Lemma 2)}$$

$$= \frac{-\gamma\, \mathbb{E}\Big[\hat{f}_W^{\hat{s}_{i_0}}(x')\Big]}{\mathbb{E}[\,W(X, x')\,]}.$$

Using the finite expansion of the fitted RKHS component,

$$\hat{f}_W^{\hat{s}_{i_0}}(x') = \hat{\beta}_{\hat{s}_{i_0}} + \frac{1}{\gamma} \sum_{i\in\mathcal{D}_{\text{calib}}\cup\{i_0\}} \hat{v}_{\hat{s}_{i_0},i}\, W(X_i, x'),$$

we can further write

$$\mathbb{E}\Big[\mathbf{1}\Big\{\mathcal{L}_\lambda\big(\mathcal{S}(\widehat{\mathbf{A}}_{i_0}; s_{i_0}), \mathbf{A}_{i_0}\big) \leq \rho\Big\} - (1-\alpha) \,\Big|\, X'_{i_0} = x'\Big] = \frac{-\mathbb{E}\Big[\sum_{i\in\mathcal{D}_{\text{calib}}\cup\{i_0\}} \hat{v}_{\hat{s}_{i_0},i}\, W(X_i, x')\Big]}{\mathbb{E}[\,W(X, x')\,]},$$

which completes the localized reweighting conformal prediction.

# D EXPERIMENTAL DETAILS

## D.1 CLINICAL TEXT CLASSIFICATION

This dataset consists of natural language descriptions of symptoms annotated with 22 corresponding diagnoses (Gretel AI, 2024). In total, it contains 1,065 English-language symptom descriptions, of which 853 (80%) are allocated for training and 212 (20%) for testing.

As described in the main text, each training symptom description is extended with five additional sentences using `GPT-4.1 nano` with temperature 1.5. Each augmented sentence is paired with the original label and treated as a new sample. To assess quality, we applied `Gemini-2.5-Pro` to generations from 100 randomly selected documents (yielding 500 new samples) and `Gemini-2.5-Flash` to generations from all 853 training documents (yielding 4,265 new samples).

For evaluating augmentation methods, we fixed a set of 100 documents with scores from Gemini-2.5-pro and randomly sampled an additional 400 documents from the training set. From these 500 documents, we applied different filtering strategies. In particular, for CondCP, we applied Latent Dirichlet Allocation (LDA) with 18 latent mixtures to the entire training set, where the number of mixtures was chosen based on log-likelihood validation, in order to estimate the latent mixture representation of each document. The procedure was repeated 20 times, and we reported precision, recall, and accuracy.

**Prompt for Data Generation**

```
You are given a description of a disease.

Description: {symptom}

Task: Extend the symptom description with additional details that
still plausibly describe the SAME disease.
- Write EXACTLY 5 sentences.
- Do not copy wording from the original; paraphrase and add
plausible details consistent with the same condition.
- Avoid lists, bullets, headings, or numbering; just 5 full
sentences in a single paragraph.
- No disclaimers, no citations, no markdown.
```

**Prompt for Evaluation**

```
You are evaluating individual symptom descriptions for diseases.

Scoring instructions:
- Assign each description a score between 0 and 1, rounded to two
decimal places.
- Criteria: The description should plausibly match the specified
disease and avoid confusion with other diseases.
- Use the full 0-1 range: 1 = perfectly clear, specific, and
accurate; 0 = completely unusable.
- 0.5 is the threshold: any description with a score <= 0.5 should
be dropped to prevent misclassification.

For reference, here is the complete list of possible diseases:
{disease_ls}

Output requirements:
- Output only the scores, one per line, in the same order as the
input  cases.
- Do not include explanations, text, or formatting other than the
numeric scores.

Case : Disease: {diag} : Symptom: {symp}
```

### D.2 ABSTRACT TOPIC CLASSIFICATION

ArXiv hosts more than 1.5 million articles across diverse fields. For this analysis, we use a random sample of 1,000 abstracts published after January 1, 2021, distributed evenly across five statistical categories: statistical methodology, statistical machine learning, statistical applications, statistical computation, and statistical theory (Artgor, 2019). The classification task is challenging because these categories are closely related. We split the 1,000 abstracts into 800 for training and 200 for testing.

Each training abstract is extended with six additional sentences using GPT-4.1 nano with temperature 1.5. Every two consecutive sentences are grouped as a new sample, paired with the label

of the original abstract. To assess quality, we applied `Gemini-2.5-Pro` to generations from 100 randomly selected abstracts (yielding 300 new samples) and `Gemini-2.5-Flash` to generations from all 800 training abstracts (yielding 2,400 new samples). For each abstract $X_i$, with extended groups $\{G_{ik}\}_{k=1}^{3}$, Gemini-pro scores $\{A_{ik}\}$, and Gemini-flash scores $\{\hat{A}_{ik}\}$, we define

$$S(\hat{\mathbf{A}}_i, \mathbf{A}_i) = \inf\left\{\tau : \left|\{G_{ik} : 1 \le k \le 3,\, \hat{A}_{ik} \ge \tau,\, A_{ik} < 0.5\}\right| = 0\right\}.$$

The evaluation procedure follows the same protocol as in clinical text classification. We fixed a set of 100 documents with scores from `Gemini-2.5-Pro` and randomly sampled an additional 200 documents from the training set. From these 300 documents, we applied different filtering strategies. Latent Dirichlet Allocation (LDA) was then performed with 5 latent mixtures, consistent with the number of categories in the dataset. Just as the clinical text example, we fine-tune a small classifier (`distilbert-base-uncased`) for topic prediction.

**Prompt for Data Generation**

```
You are given a statistical abstract.

Abstract: {abstract}

Task: Extend the abstract with additional details that remain
consistent with the SAME statistical topic.
- Write EXACTLY 6 sentences.
- Do not copy wording from the original; paraphrase and add
plausible extensions consistent with the same subject.
- Avoid lists, bullets, headings, or numbering; just 6 full
sentences in a single paragraph.
- No disclaimers, no citations, no markdown.
```

**Prompt for Evaluation**

```
You are evaluating individual sentences from extended statistical
abstracts.

Scoring instructions:
- Assign each sentence a score between 0 and 1, rounded to two
decimal places.
- Criteria: The sentence should plausibly match the specified topic,
remain coherent, and avoid drifting into other topics from the list.
- Use the full 0-1 range: 1 = perfectly clear, on-topic, and
informative; 0 = completely unusable.
- 0.5 is the threshold: any sentence with a score <= 0.5 should be
dropped to prevent topic drift.

Output requirements:
- Output only the scores, one per line, in the same order as the
input cases.
- Do not include explanations, text, or formatting other than the
numeric scores.

Case : Topic: {topic} : Sentences: {sent}
```

### D.3 TWITTER MESSAGE SENTIMENT ANALYSIS

The dataset (Kadara, 2018) contains text segments from Twitter messages, each labeled with the predominant emotion expressed. The emotions are categorized into six classes: sadness, joy, love, anger, fear, and surprise. We randomly sampled 1,200 messages, evenly distributed across the six categories, and split them into 1,000 for training and 200 for testing.

Each training message was extended with five additional sentences using `GPT-4.1 nano` with temperature 1.5, with each sentence paired to the original label as a new sample. For evaluation, `Gemini-2.5-Pro` scored generations from 100 documents (500 samples), while `Gemini-2.5-Flash` covered all 1,000 training documents (5,000 samples). The evaluation procedure follows the same protocol as in clinical text classification: we fixed a set of 100 documents with scores from `Gemini-2.5-Pro` and randomly sampled an additional 200 documents from the training set. We define the non-conformity score as

$$S(\hat{\mathbf{A}}_i, \mathbf{A}_i) = \inf \left\{ \tau : \left| \{ G_{ik} : 1 \leq k \leq 5, \, \hat{A}_{ik} \geq \tau, \, A_{ik} < 0.5 \} \right| \leq 1 \right\}.$$

The remaining steps were identical to the previous cases, except that here we applied LDA with six mixture components.

**Prompt for Data Generation**

```
You are given a short Twitter message.

Message: {tweet}

Task: Extend the message with additional content that preserves the
SAME sentiment and topic.
- Write EXACTLY 5 sentences.
- Paraphrase and expand naturally; do not copy wording from the
original.
- Vary phrasing, tone, and detail while remaining consistent with
the sentiment.
- Avoid lists, bullets, hashtags, mentions, links, or numbering;
produce 5 full sentences in a single paragraph.
- No disclaimers, citations, or markdown.
```

**Prompt for Evaluation**

```
You are evaluating individual sentences for sentiment consistency.

Scoring instructions:
- Assign each sentence a score between 0 and 1, rounded to two
decimal places.
- Criteria: The sentence should clearly reflect the SPECIFIED
sentiment, remain coherent, and avoid conflicting emotions.
- Use the full 0-1 range: 1 = perfectly consistent and natural;
0 = completely unusable.
- 0.5 is the threshold: any sentence with a score <= 0.5 should
be excluded.

Output requirements:
- Output only the scores, one per line, in the same order as the
input cases.
- Do not include explanations, text, or formatting beyond the
numeric scores.

Case: Sentiment: {senti} Sentence: {sent}
```

D.4 DIVERSITY OF AUGMENTED TEXT

**Diversity of Selected Augmentations.** To evaluate the diversity of the augmentation techniques, we compute their Shannon entropy. The results are reported in Table 2 for the Diagnosis, Abstract and Sentiment Datasets. Overall, we find that augmentation generally increases the diversity of the training data (e.g., the Diagnosis dataset features an unaugmented diversity of 6.92, compared to 8.02 for the CondCP filter and 8.52 for the unfiltered dataset). The filtered versions typically show

lower diversity compared with the unfiltered versions since poor-quality generations are excluded. We also note that the CondCP-filtered generations exhibit lower diversity than those filtered by other algorithms. This outcome is expected, given the nature of conformal prediction. Nonetheless, the reduction in diversity is relatively small, highlighting that the CondCP filter effectively preserves the essential information contained in the training data.

**Sensitivity to the choice of $\rho$ and $\lambda$:** To assess the sensitivity of the proposed CondCP approach to the choice of hyperparameters (e.g. $\rho$ and $\lambda$), we report the Shannon entropy of the CondCP-filtered results under different hyperparameter configurations in Table 3. Overall, larger values of $\lambda$ and smaller values of $\rho$ tend to reduce diversity. This highlights the fact that these hyperparameters should be chosen to balance diversity against faithfulness to the original data distribution. Their sensitivity depends on the dataset as well as on both the gold-standard and surrogate diversity measures.

| Data | Unaugmented | Unfiltered | Flash Filter | Hybrid Filter | CondCP Filter |
|---|---|---|---|---|---|
| Diagnosis | 6.92 | 8.52 | 8.14 | 8.14 | 8.02 |
| Abstract | 9.64 | 9.83 | 9.81 | 9.81 | 9.76 |
| Sentiment Analysis | 8.69 | 9.34 | 9.23 | 9.23 | 8.96 |

Table 2: Shannon entropy of augmented data across datasets under different augmentation methods

| Diagnosis | | | | | | |
|---|---|---|---|---|---|---|
| | $\lambda = 0.3$ | $\lambda = 0.4$ | $\lambda = 0.5$ | $\lambda = 0.6$ | $\lambda = 0.7$ | $\lambda = 0.8$ |
| $\rho = 0$ | 7.67 | 7.53 | 7.42 | 7.44 | 7.36 | 7.32 |
| $\rho = 1$ | 8.14 | 8.10 | 8.02 | 7.94 | 7.87 | 7.81 |
| $\rho = 2$ | 8.19 | 8.16 | 8.13 | 8.13 | 8.05 | 8.03 |
| Abstract | | | | | | |
| | $\lambda = 0.3$ | $\lambda = 0.4$ | $\lambda = 0.5$ | $\lambda = 0.6$ | $\lambda = 0.7$ | $\lambda = 0.8$ |
| $\rho = 0$ | 9.76 | 9.76 | 9.76 | 9.76 | 9.76 | 9.76 |
| $\rho = 1$ | 9.77 | 9.77 | 9.77 | 9.77 | 9.76 | 9.76 |
| $\rho = 2$ | 9.77 | 9.77 | 9.77 | 9.77 | 9.76 | 9.76 |
| Sentiment Analysis | | | | | | |
| | $\lambda = 0.3$ | $\lambda = 0.4$ | $\lambda = 0.5$ | $\lambda = 0.6$ | $\lambda = 0.7$ | $\lambda = 0.8$ |
| $\rho = 0$ | 8.99 | 8.94 | 8.89 | 8.90 | 8.89 | 8.87 |
| $\rho = 1$ | 9.08 | 9.01 | 8.96 | 8.96 | 8.94 | 9.01 |
| $\rho = 2$ | 9.08 | 9.07 | 9.09 | 8.99 | 8.98 | 8.97 |

Table 3: Shannon entropy of CondCP filtered data across different $\lambda$ and $\rho$

## D.5 COMPARISON BETWEEN GEMINI-2.5-PRO SCORES AND GEMINI-2.5-FLASH SCORES

Figure 5 presents a comparison of evaluation scores between Gemini-2.5-Pro and Gemini-2.5-Flash across datasets. While the two scores show a clear positive correlation, they are not perfectly aligned.

## D.6 EXPERIMENTAL RESULTS FOR TEXT DATA UNDER LOW-TEMPERATURE GENERATION

The LLM-augmented experiments presented in the main text were conducted with a high generation temperature of 1.5. For completeness, we report in Figure 6 the corresponding results obtained under a low-temperature setting with generation temperature 0.3. As anticipated, CondCP filter offers limited improvement in this regime due to the substantially reduced diversity of generated samples: at low temperature, the LLM predominantly produces highly frequent or canonical outputs, leaving little variation for the filtering mechanism to act upon. Consequently, overall performance is worse than the high-temperature setting with CondCP filtering reported in the main text.

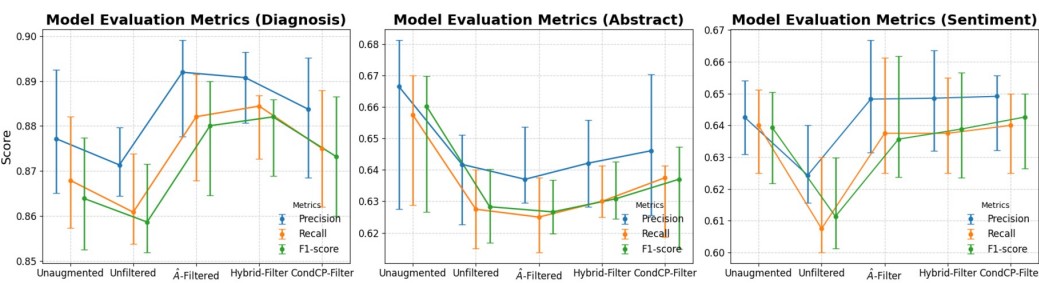

Figure 5: Scatter plots comparing Gemini-Pro and Gemini-Flash scores for symptom descriptions, statistical abstracts, and Twitter messages datasets.,

Figure 6: Evaluation of different data augmentation methods on diagnosis prediction, abstract topic prediction, and Twitter message sentiment prediction with generation temperature 0.3.

## D.7 ASSESSMENT OF RISK-CONTROL VIOLATIONS

For each text dataset, we partition the samples with `Gemini-2.5-pro` scores into 10 folds. In each split, we use 9 folds to train CondCP and evaluate the empirical violation rate on the remaining fold, which is the frequency with which the number of low-quality generations exceeds the allowed threshold $\rho$. The results are shown in Figure 7 , which illustrates the risk control achieved by our method.

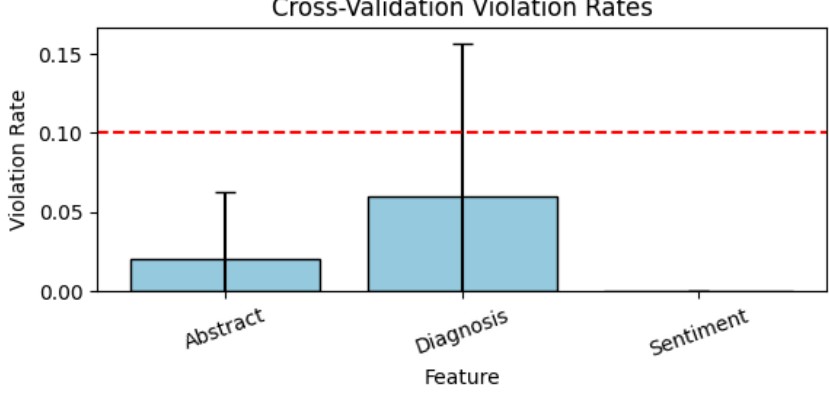

Figure 7: Empirical violation rate for allowing at most $\rho$ low quality generations. Error bars represent $\pm 1$ standard deviation.

## D.8 ADDITIONAL EXPERIMENTS ON IMBALANCED CLASSIFICATION

| Dataset | Hidden Dim | Context Dim ($c$) | Latent Dim | Epochs | Learning Rate |
|---------|-----------|-------------------|-----------|--------|---------------|
| Thyroid | 64 | 32 | 32 | 100 | $1 \times 10^{-3}$ |
| Credit Card | 64 | 32 | 32 | 100 | $1 \times 10^{-3}$ |
| MNIST-7 vs. rest | 64 | 32 | 32 | 100 | $1 \times 10^{-3}$ |

Table 4: CVAE hyperparameters across datasets. Hidden dimension refers to the encoder/decoder width, context dimension is the size of the seed-conditioned embedding $c = h_\psi(x)$, and latent dimension is the size of the stochastic latent variable $z$.

In this appendix, we offer a deep dive into some of the examples presented in this paper for the imbalanced classification results (Section 4.2).

### D.8.1 EXPERIMENT DETAILS

**Datasets.** We evaluate our framework on three benchmark datasets that cover a wide spectrum of imbalance ratios, dimensionalities, and application domains:

- **European Credit-Card Fraud** (Kaggle): 284,807 transactions with 492 frauds (0.17% positives). Each record contains transaction time, amount, and 28 PCA-compressed features (V1–V28). This dataset is widely used as a canonical benchmark for extreme class imbalance.

- **Thyroid Disease** (OpenML-38): 2,644 patient records with 6.4% positives. Features include demographic covariates, hormone levels, and binary medical indicators. This dataset represents a typical medical diagnosis problem with moderate imbalance.

- **MNIST-7 vs. Rest** (OpenML-554): 70,000 handwritten digits recast into a binary task of distinguishing "7" (10.9% positives) from all other digits. While less imbalanced, this high-dimensional vision-like dataset provides a contrasting baseline where signal is strong and plentiful.

**Conditional VAE (CVAE).** We generate minority samples with a Conditional Variational Autoencoder (CVAE) (Kingma & Welling, 2013; Sohn et al., 2015) that is *conditioned on an actual minority seed*. Let $h_\psi : \mathbb{R}^d \to \mathbb{R}^c$ be a small *context net* that maps a reference minority instance $x$ to a context vector $c = h_\psi(x)$. The encoder receives the concatenation $[x, c]$ and outputs a Gaussian $q_\phi(z \mid x, c) = \mathcal{N}(\mu_\phi(x, c), \mathrm{diag}(\sigma_\phi^2(x, c)))$; the decoder reconstructs $x$ from $[z, c]$ via $p_\theta(x \mid z, c)$. Both encoder and decoder are two-layer MLPs with ReLU activation.

We train on minority data only ($y = 1$) with the ELBO using Adam optimizer for 100+ epochs (features are MinMax-scaled). At generation time, given a seed $x_s$ we compute $c_s = h_\psi(x_s)$ and draw K candidates by sampling $z_k \sim \mathcal{N}(0, \tau^2 I)$ (with $\tau > 0$ and decoding $g_k = g_\theta(z_k, c_s)$. This *seed-conditional* design produces local, seed-aware variations that stay on the minority manifold with controlled dispersion $\tau$. The raw candidates are then quality-scored and filtered by our $\widehat{A}$ regressor and conformal thresholds before being added for training a classifier. See the detailed choice of architecture in Table 4.

**Conformal Prediction conditioned on Latent Representation** We apply conditional conformal filtering that operates in a learned latent representation of the data (Anonymous, 2026). Specifically, we project the feature space into a lower-dimensional latent embedding using Principal Component Analysis (PCA) before applying the conformal calibration step. For each dataset, we tune the latent dimension to reflect its scale: 2 for *Thyroid* and *Credit Card Fraud*, and 16 for *MNIST-7*.

### D.8.2 EVALUATING THE QUALITY OF THE (SELECTED) GENERATIONS

We examine the MNIST 7 example, an imbalanced classification task where MNIST digits are classified as 7 or not 7, with 7s being underrepresented. We evaluate the effect of temperature and selection on the diversity of the generated samples, as the results in this example should be easily interpretable visually.

**Understanding the effect of temperature on the diversity of the samples.** Figure 8 highlights a few examples of generations of the digit 7 for different temperatures $\tau$. In low temperature settings (e.g. $\tau = 0.1$), the model generates almost identical samples. In moderately high temperature settings ($\tau = 2$), the model starts to generate more variable shapes of the digit 7. However, as the temperature becomes too high ($\tau = 10$), the synthetic data become extremely noisy.

Figure 9 and 10 further illustrate the temperature effect through principal component analysis, comparing real and generated data. As the temperature ($\tau$) increases, the synthetic samples gradually explore a wider area of the real data. However, excessively high temperatures (e.g., $\tau = 10$) cause the generator to sample outside the MNIST distribution, resulting in points that do not align with the original data's structure.

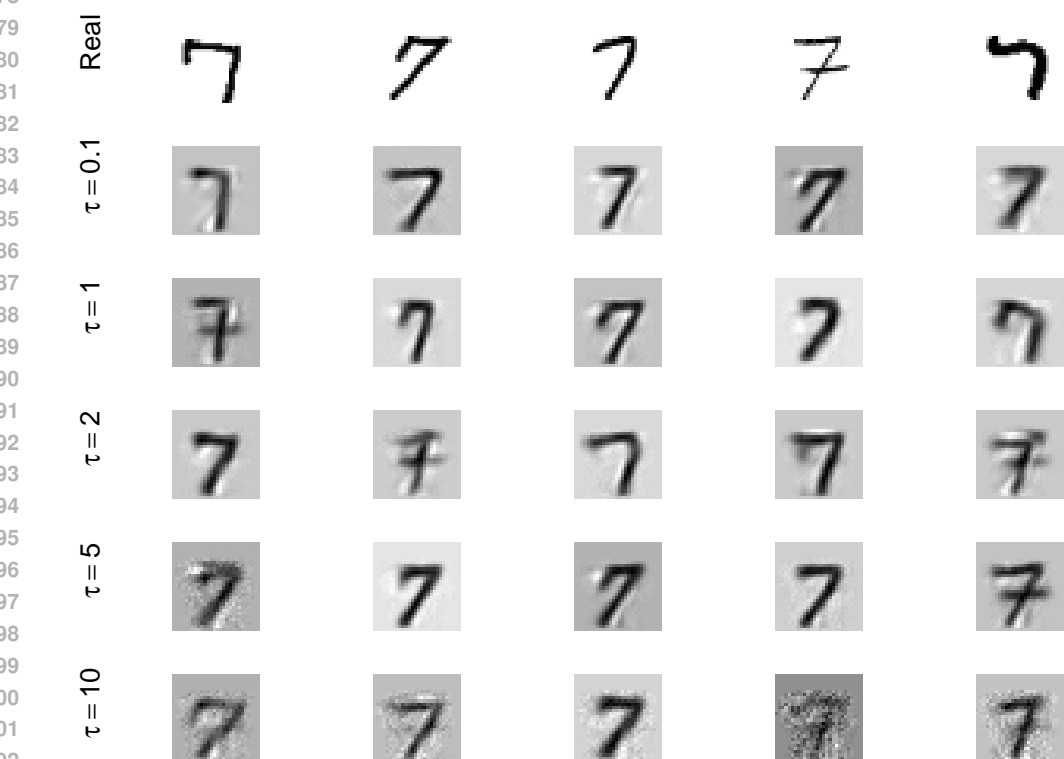

Figure 8: Generated minority digit (7) by the VAE model for varying temperature ($\tau$).

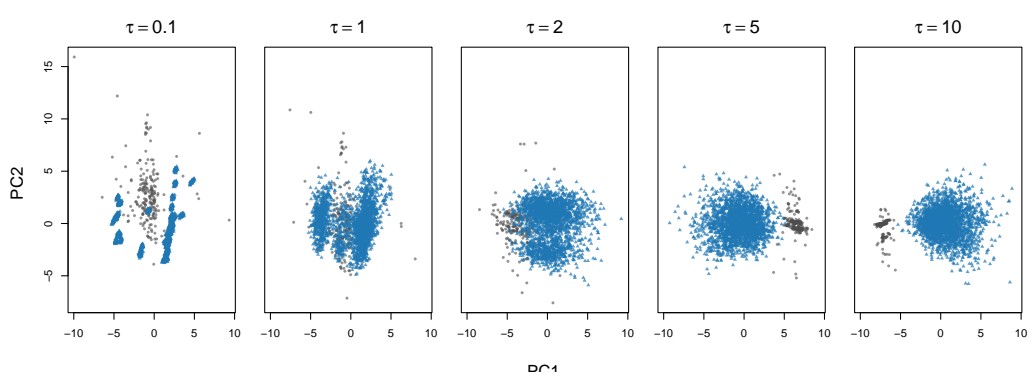

Figure 10: PCA visualization of real samples from thyroid dataset and VAE-generated samples under different temperature values $\tau$. The gray circle points denote the real data, and the blue triangular points denote the generated data.

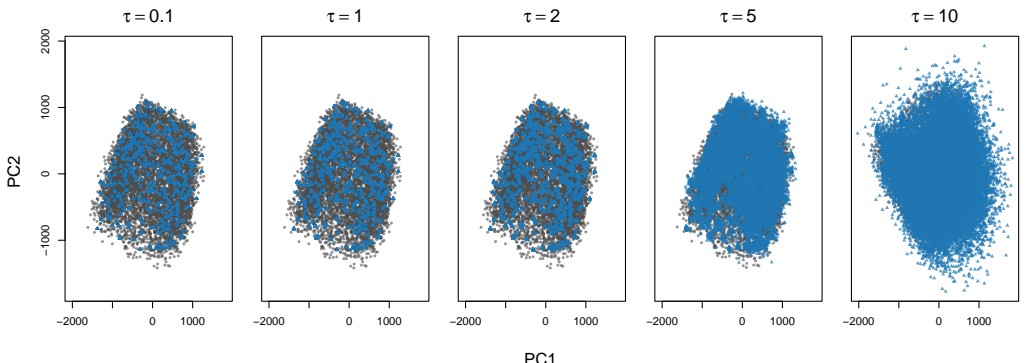

Figure 9: PCA visualization of real MNIST-7 digits and VAE-generated samples under different temperature values $\tau$. The gray circle points denote the real data, and the blue triangular points denote the generated data.

To quantify these effects and directly examine the role of temperature, we measure the diversity of the (unfiltered) generated samples for different values of the parameter $\tau$ using the stable rank. The results, shown in Table 5, confirm that increasing the sampling temperature produces higher-diversity synthetic data; however, but this diversity is uncontrolled. Beyond moderate temperatures, the generator begins to sample outside the real data manifold, producing overly noisy or implausible examples. Figure 8 illustrates this phenomenon, where high-temperature samples turn out to be overly noisy. This supports the motivation stated in the introduction: while higher temperatures can, in principle, expose rare modes, naively relying on high-temperature augmentation is harmful because it injects low-quality, out-of-support samples.

| $\tau$ | Thyroid | Credit Card Fraud | MNIST 7 |
|---|---|---|---|
| 0.1 | $6.072 \pm 0.689$ | $1.960 \pm 0.039$ | $12.048 \pm 0.754$ |
| 1 | $8.238 \pm 0.773$ | $1.989 \pm 0.034$ | $12.020 \pm 0.745$ |
| 2 | $8.026 \pm 1.075$ | $1.998 \pm 0.095$ | $12.125 \pm 0.707$ |

Table 5: Diversity measure by Stable Rank of unfiltered augmentation with varying temperature ($\tau$). Mean and standard deviation computed across different splits (seed). The smaller $\tau$ will lead to generate the samples closer to the existing point, and the higher $\tau$ will lead to more noisy generation.

**Understanding the effect of selection.** Figures 11 and 12 show examples of accepted and rejected samples, respectively. As expected, we see that the rejected samples feature more low quality (extremely blurry and jagged) sevens, compared to the selected ones: the condCP selection seems to select samples that are more realistic.

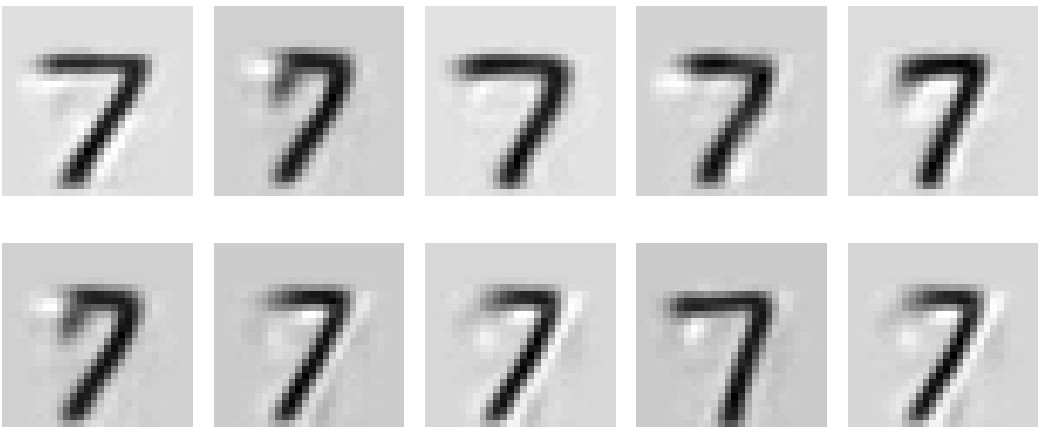

Figure 11: Examples of accepted generations by CondCP with $\tau = 0.1, \lambda = 0.5, \rho = 2$.

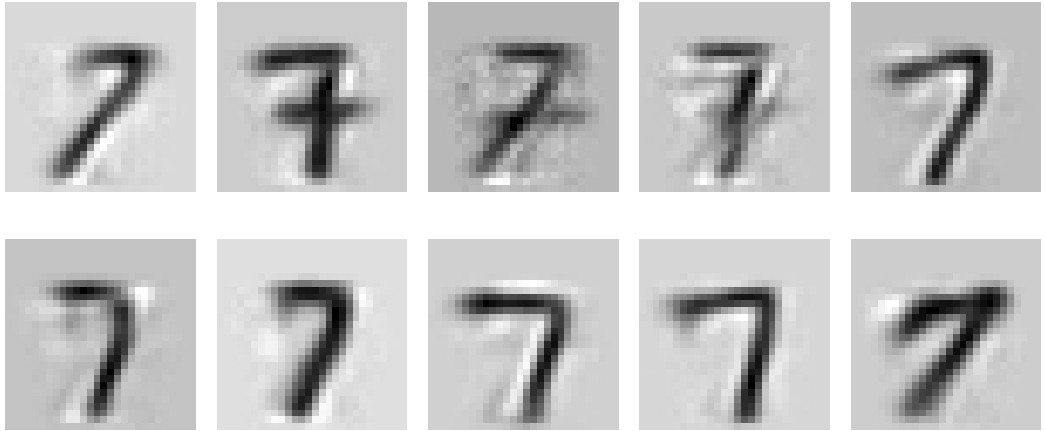

Figure 12: Examples of generations filtered out by CondCP with $\tau = 0.1, \lambda = 0.5, \rho = 2$.

This effect extends to other datasets. In Figure 13, we visualize the effect of the selection on the Thyroid dataset. To this end, we first extracted important predictors using regression on training dataset, and the three variables, `on thyroxine` (binary treatment indicator), `T3` (serum triiodothyronine level), and `TT4` (total thyroxine level), emerged as significant predictors. The plots show, for each variable and each filtering strength $\lambda$, how the distribution of accepted synthetic samples aligns with the real minority-class distribution. Across features, accepted samples (blue) consistently match the true minority distribution (black) better than rejected samples (gray), demonstrating that the filtering criterion preferentially retains synthetic points that lie on the true data-support for the minority class.

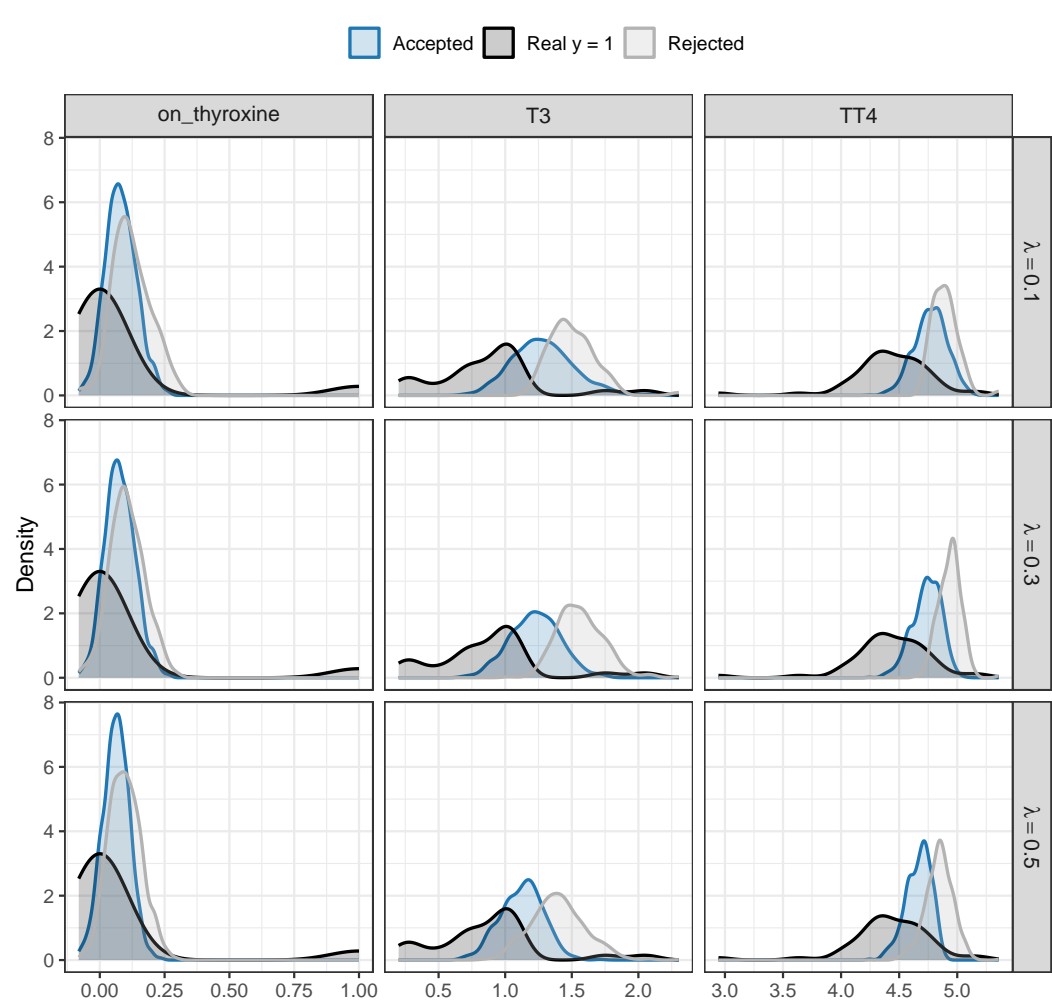

Figure 13: Using the thyroid dataset, we first fit a simple logistic regression on 60% of the real data to predict the minority thyroid-disease class. Three variables, `on thyroxine` (binary treatment indicator), `T3` (serum triiodothyronine level), and `TT4` (total thyroxine level), emerged as significant predictors. The tolerance parameter $\rho$ is fixed to be 2.

