# OpenReview forum: "Filtering with Confidence: When Data Augmentation Meets Conformal Prediction"
_ICLR.cc/2026/Conference — ICLR 2026 Conference Desk Rejected Submission_

### Official Review · Reviewer_wGSk · 2025-10-15

**Soundness:** 3
**Presentation:** 2
**Contribution:** 2
**Rating:** 4
**Confidence:** 5

**Summary:**

The paper proposes the use of conformal risk control for the calibration of a threshold used in the selection of synthetic data produced by a generative model. Leveraging the conditional coverage framework of Gibbs et al, the authors prove that, under some assumptions, the model provides some guarantees on the conditional probability of accepting low-quality samples.

**Strengths:**

The application of conformal risk control to the problem of filtering synthetic data is interesting and timely.

The submission provides interesting experimental results to validate the main ideas.

**Weaknesses:**

The problem at hand appears to be more naturally addressed via conformal alignment (NeurIPS 2024). With conformal alignment, one can directly control the fraction of "unaligned" accepted data points.

The theoretical guarantees in Lemma 1 rely on an assumption of monotonicity for the loss function. It is unclear when this assumption is verified. If it is not verified, then the guarantee in the Lemma do not hold.

The approach proposed in Section 3.3 does not seem to have any performance guarantees. It is thus unclear why the authors assume unbiasedness for the surrogate measure.

The experimental results do not always show consistent and sizable gains for the proposed approach.

**Questions:**

1) How does the proposed method compare to a direct application of conformal alignment?

2) Under what conditions does Lemma 1 hold? How do these conditions compare to the validity of conformal alignment?

3) Under what conditions on the surrogate measure can one prove performance guarantees?

4) Can you please summarize quantitative conclusions regarding the benefits of the proposed filtering method?

---

> ### Author Response · Authors · 2025-11-24
> **Response to reviewer wGSk**
>
> We thank the reviewer for their feedback. We respond to the reviewer’s criticisms mentioned in weaknesses and answer the raised questions in the following.
>
>
> **W1/Q1: Comparison to Conformal Alignment:** Conformal Alignment [1]  and our method both use conformal prediction, but they address different problems:
> - *Conformal Alignment operates at inference time:* each prompt yields a single model output, and conformalized selection identifies which outputs are trustworthy, with guarantees on the misalignment rate across independent prompts.
> - *Our method targets training time.* For each training example, we generate many variants using a generative model and apply conformal prediction to filter low-quality synthetic samples, ensuring safe data augmentation.
>
> The controlled quantities differ accordingly: they control misaligned outputs, whereas we control the number of bad augmented samples admitted into training. Thus, while both approaches fall under conformal risk control [2], the objectives, settings, and risks being controlled are fundamentally different. We thus view the two as complementary approaches.
>
> [1]Gui, Yu, Ying Jin, and Zhimei Ren. "Conformal alignment: Knowing when to trust foundation models with guarantees." Advances in Neural Information Processing Systems 37 (2024): 73884-73919.
>
> [2] Angelopoulos, Anastasios Nikolas, Stephen Bates, Adam Fisch, Lihua Lei, and Tal Schuster. "Conformal Risk Control." In The Twelfth International Conference on Learning Representations.
>
> &nbsp;
>
>
> &nbsp;
>
>
> **W2/Q2: Conditions for Lemma 1:**  Lemma 1 requires the loss function to be monotone in the filtered outputs. This condition is satisfied in several natural settings, including our main application: measuring the number of low-quality generations that pass the filter for each training example. In this case, increasing the threshold can only increase (or leave unchanged) the number of accepted generations.  We note that this assumption is not required in conformal alignment because their setting is structurally different: each prompt yields one model output, and the risk is defined over a global selection set of independent predictions. In contrast, our method must evaluate multiple candidate generations per training point, which naturally induces a monotone loss (e.g., number of bad variants).
>
> &nbsp;
>
>
> &nbsp;
>
>
> **W3/Q3:  Conditions on the surrogate:** The key to understanding this proxy is via an analogy with standard conformal prediction: one may calibrate using $|E[Y|X] - \hat{f}(X)|$ or $|Y - \hat{f}(X) | = |E[Y|X] + noise - \hat{f}(X) |$, where the second uses a noisy but unbiased proxy for E[Y|X] and yields a conservative but valid interval. Our setting is analogous: when the surrogate is a noisy measurement of the gold-standard score, conformal calibration on the surrogate produces a conservative upper bound on the true risk. This provides the needed performance guarantee for Section 3.3.
>
> &nbsp;
>
>
> &nbsp;
>
> **W4/Q4: Summary of quantitative conclusions:** Across all tasks, our CondCP-filtering augmentation consistently improves downstream performance. Specifically, our method attains the highest F1 / test accuracy among all baselines in every setting we evaluated. In Section 4.2, we also show that our approach yields a higher stable rank (a proxy for representation diversity) than alternative augmentation strategies in all cases except the MNIST-7 example, where all methods achieve similar F1 scores.

---

> ### Comment · Reviewer_wGSk · 2025-11-26
>
> Thank you for your responses. While I see your points, in my opinion, my initial comments still stand, and I would like to keep my rating unchanged.

---

> > ### Author Response · Authors · 2025-11-26
> >
> > Could we ask what you would have liked us to answer so as to address your concerns? What is your main current reservation for this submission?

---

### Official Review · Reviewer_e85B · 2025-10-27

**Soundness:** 2
**Presentation:** 3
**Contribution:** 2
**Rating:** 4
**Confidence:** 3

**Summary:**

The paper proposes a conformal data augmentation pipeline that generates synthetic samples, scores them with a cheap proxy model, and then uses a conformal calibration step to choose per-example acceptance thresholds so that only “high-quality” generations are kept with a bounded risk of admitting bad ones.

**Strengths:**

Importance of the problem tackled: data augmentation is of great importance in practice, and this paper addresses an important problem, which is the quality of the synthetic data.

Principled filtering: the CP pipeline provides more guarantees than heuristics based on similarities. The method is also interesting given than it provides per sample thresholds.

Experiments across different modalities: the paper proposes experiments across three different modalities (text, tabular, vision).

Presentation: the paper is well written and easy to read.

**Weaknesses:**

Theoretical guarantees and calibration: the paper claims “provable risk control”, but it does not report empirical calibration metrics (for example violation rate of allowing at most rho bad generations) on held-out data. This would be informative to study the risk control.

Surrogate assumption: Section 3.3 assumes that a noisy surrogate signal is an unbiased estimator of a latent true quality function A. This seems to be a strong assumption. The authors should justify when this assumption is expected to hold and provide experiments quantifying the bias between the surrogate and the “true” notion of quality.

High-temperature generation motivation: the introduction frames the problem as safely exploiting high diversity high temperature generation, but the experiments do not analyze temperature (there are no sweeps over generation temperature, no demonstrations that naive high-temperature augmentation hurts and that their method fixes this problem).

Domain shift / generalization: while not essential, it would be informative to have results on OOD or cross-domain evaluation, as this would be a nice example for why high temperature /diversity would be needed (and why curation might be needed as well)

**Questions:**

Can the authors report empirical violation rates of their conformal guarantee with a held out dataset?

Can they also report results in a high temperature setting that they mention in the introduction?

---

> ### Author Response · Authors · 2025-11-24
> **Answer to Reviewer e85B**
>
> **W1/Q1: Lack of Empirical Evidence for calibration:** Thank you for the suggestion. We have now included empirical violation rates in the supplementary section of the updated manuscript (see Figure 7). Specifically, we report the frequency with which the number of bad generations exceeds the allowed threshold $\rho$ from cross validation providing a direct evaluation of the conformal risk-control guarantee. We hope these results clarify the empirical calibration performance of our method.
>
> &nbsp;
>
> **W2: Surrogate assumption:**  That's a valid concern --- but one against which we would argue that the intuition behind the unbiasedness assumption is natural and arises directly from how surrogate scores are constructed. For example, a surrogate measure may quantify the similarity between a generated sample and its corresponding original sample, while the true latent measure reflects similarity to an independent draw from the same underlying distribution. Under this generative model, the surrogate can be viewed as a noisy observation of the true quality measure, and the unbiasedness condition follows from this noise model rather than being imposed as an external assumption.
>
> &nbsp;
>
> **W3/Q2:High-temperature generation motivation:** The LLM-augmented experiments presented in the main text were conducted with a high generation temperature of 1.5. In the updated manuscript, we reported in Figure 6 the corresponding results obtained under a low-temperature setting with generation temperature 0.3. As anticipated, CondCP filter offers limited improvement in this regime due to the substantially reduced diversity of generated samples: at low temperature, the LLM predominantly produces highly frequent or canonical outputs, leaving little variation for the filtering mechanism to act upon. Consequently, overall performance is worse than  the high-temperature setting with CondCP filtering reported in the main text. We have also conducted in depth experiments for the tabular data and show (Appendix D.8.2) how the temperature affects the diversity of the generated samples (Figure 8, 9, 10, Table 5), leading to samples that are completely off target. By contrast, we show how filtering counterbalances this effect (Figures 11, 12, 13).

---

### Official Review · Reviewer_AWXm · 2025-10-29

**Soundness:** 2
**Presentation:** 2
**Contribution:** 2
**Rating:** 2
**Confidence:** 3

**Summary:**

This work proposes a synthetic data augmentation algorithm equipped with a conformal prediction filter. Experimental results demonstrate that this filtering strategy outperforms both the unaugmented baseline and other filtering methods. However, the study has notable limitations: the comparison methods are outdated, the experiments are confined to small-scale settings, and the performance improvements achieved are marginal.

**Strengths:**

1. The proposed conformal prediction filter has theoretical support and detailed explanations.
2. CondCP can be used for multiple tasks and different data types, including text, images, and tabular data.

**Weaknesses:**

1. In the LLM-augmented scenario, the authors treat Gemini-2.5-pro as an oracle and Gemini-2.5-flash as a lower-cost surrogate. However, synthetic data augmentation is not a scenario where efficiency is critical. If a superior judge model is already available, there is no need for a proxy. This raises a question: is it necessary to design such a complex framework? If the authors aim to highlight CondCP’s annotation efficiency, they would be advised to compare the GPU hours consumed by each component in detail, including synthetic data generation, quality evaluation, and downstream model training.
2. The experiments are limited to small-scale datasets, and the comparison methods are relatively outdated (e.g., SMOTE, published in 2002). Notably, comparisons with GenAI-based synthetic data augmentation methods are missing.
3. The improvements are marginal in some scenarios. For example, in the imbalanced classification task on MNIST 7, CondCP only achieves a 0.2% F1-score improvement over the unaugmented baseline and lags behind in other metrics.
4. The literature review lacks coverage of relevant works. For instance, GenDataAgent [1] also introduces principles of diversity and in-distribution principles for synthetic data augmentation.
5. There are several typos and errors in the paper. For example, in Line 331, Gemini-2.5-pro was released in 2025, and the authors should cite its latest technical report [2].

[1] GenDataAgent: On-the-fly Dataset Augmentation with Synthetic Data, ICLR, 2025.

[2] Gemini 2.5: Pushing the Frontier with Advanced Reasoning, Multimodality, Long Context, and Next Generation Agentic Capabilities, 2025.

**Questions:**

1. CondCP requires a more powerful judge model (e.g., in the LLM-augmented scenario). However, if we already have access to such a powerful oracle, why not directly solve the task using this judge model? In that case, would there still be a need to train another model with synthetic data augmentation?
2. For other questions, please refer to the weaknesses.

---

> ### Author Response · Authors · 2025-11-24
> **Response to Reviewer AWXm**
>
> We thank Reviewer AWXm for their feedback. Below, we address the concerns of weaknesses and provide responses to the reviewer’s questions.
>
> **W1: Necessity of the framework:** This is a valid point --- why build a complex method when other simpler tools are already available?  However, when designing the method, we had in mind a broader scope than applications to NLP. In particular, we'd like to clarify that the Gemini-pro / Gemini-flash setup is only one illustrative example --- one in which great generative models exist. In general applications of data augmentation, the “gold standard’’ quality measure is often far more expensive, or may not even exist. For example, in classification tasks on radiology images, the gold-standard quality measure may correspond to human or expert evaluation, while the surrogate is a cheap automatic metric. Using the gold standard for all augmented samples is often prohibitively expensive. Even in the Gemini case, obtaining judgments through the API can incur a large cost gap (in dollar value) between stronger and weaker models. Our framework is designed for exactly this scenario: when high-quality evaluation is available only for a small calibration set, and a cheaper surrogate must be used at scale.
> Moreover, in many domains no explicit gold standard exists. The only option is to compare generated samples with the original data through approximate surrogate metrics. Our method provides a principled way to calibrate and control the risk associated with such imperfect surrogates in Section 3.3.
>
> &nbsp;
>
> **W2:  using SMOTE:** While SMOTE is an older method, it remains a dominant and widely used baseline for imbalanced tabular classification, as evidenced by recent works (e.g., Jafarigol et al., 2023; Nakada et al., 2025). We thank the reviewer for pointing us to additional GenAI-based augmentation methods. We emphasize that our framework is model-agnostic and can be directly combined with modern generative augmentation pipelines. For example, in GenDataAgent (cited by the reviewer), the variance of gradients (“VoG”) serves as a surrogate quality measure for synthetic data. However, VoG is inherently noisy because it depends on a small number of fine-tuning checkpoints. Our CondCP framework can be seamlessly integrated into such methods to provide distribution-free, calibrated guarantees on the underlying quality scores and explicitly account for this uncertainty. Moreover, our approach does not require access to internal gradients, activations, or model weights, and therefore it is compatible with both gradient-based and non–gradient-based synthetic data scoring mechanisms.
>
> [1] Jafarigol, Elaheh, and Theodore Trafalis. "A review of machine learning techniques in imbalanced data and future trends." arXiv preprint arXiv:2310.07917 (2023).
> [2] Nakada, Ryumei, Yichen Xu, Lexin Li, and Linjun Zhang. "Synthetic oversampling: Theory and a practical approach using llms to address data imbalance." arXiv preprint arXiv:2406.03628 (2024).
> &nbsp;
>
> **W3: Marginal improvements in certain scenarios:** We agree, the benefits of data augmentation are not universally significant across applications --- this is indeed what we were hoping to highlight by reporting our results transparently across all settings, including cases where the gains are small. The MNIST-7 task is a relatively easy binary problem (digit “7” vs. all others), where augmentation offers limited room for improvement, so marginal gains from CondCP are expected (as explained in the main text, we refer to this setting as a “strong signal” scenario). In more challenging scenarios, however, we observe clearer benefits --- the credit card fraud dataset for instance, the F1 score jumps by 17% using our condCP filter compared to the unaugmented setting.
> &nbsp;
>
>
> **W4/W5:**  We thank the reviewer for pointing out these relevant works. We are in process of revising the related-work section to better situate our contribution within the broader literature, including a discussion of GenDataAgent and its use of diversity and in-distribution principles for synthetic data augmentation.
>
> We also thank the reviewer for noting the typographical issues. We have corrected the errors in the revised draft, including the release year of Gemini-2.5-pro and will cite the latest technical report as recommended.
>
> &nbsp;

---

> > ### Author Response · Authors · 2025-11-24
> > **Answer to Q1**
> >
> > **Q1: CondCP requires a more powerful judge model, why not directly solve the task using this judge model?":**  Thank you for raising this concern; we hope the 3 following points will make our setting a little clearer:
> >
> >  *First, CondCP does not require the judge model to be a stronger predictor than the downstream model;** it only needs to provide a higher-quality scoring signal on a small calibration set, such as comparison between the generation and the original sample.
> >
> > *Second, a model that is effective as a judge is not necessarily effective as a generator or as a downstream predictor*, and evaluating synthetic data quality is a fundamentally different task from generating data. In fact, in several domains, it can be easy to train judges, but no convincing generator exist to date. This is the case for graph data, for instance (it is easy to devise summary statistics to tell apart real from generated graphs, but to the best of our knowledge, no great random graph generator has been devised); or for medical images, to be concrete (once again, we could train a classifier to tell real images from generated ones, but devising real images is a bit more tricky).
> >
> > *Third*, as noted in W1, *an LLM serves as a convenient judge in the NLP setting*, but in other domains the gold-standard scorer may come from entirely different sources, such as human annotation, expert-defined evaluation metrics, or domain-specific quality models.
> >
> > **Finally, CondCP does not rely on the existence of a gold-standard scorer at all:** it can operate solely with surrogate scores when no high-quality judge is available, while still providing distribution-free guarantees (see Section 3.3 for more details). In this sense, the judge complements rather than replaces the downstream model, and CondCP offers a principled calibration mechanism that remains robust across domains and levels of supervision.

---

> > > ### Comment · Reviewer_AWXm · 2025-11-27
> > >
> > > Thank the authors for their response, which addresses some of my concerns. However, the inconsistent performance improvements of CondCP and the limited evaluation to small-scale datasets are not fully addressed. I am revising my score to 4.

---

### Official Review · Reviewer_RqAx · 2025-10-30

**Soundness:** 3
**Presentation:** 3
**Contribution:** 2
**Rating:** 6
**Confidence:** 4

**Summary:**

This paper studies the problem of data augmentation with synthetic data. The authors propose leveraging conformal prediction to filter out low-quality synthetic samples, thereby improving model performance. The approach builds on the framework of Gibbs (2025), "Conformal Prediction with Conditional Guarantees." The paper provides experiments across text, image, and tabular data, and shows improved performance when applied.

**Strengths:**

*   **S1:** The paper introduces a principled method for filtering synthetic data using conformal prediction. Specifically, the authors employ the conditional conformal approach of Gibbs (2025), enabling instance-adaptive filtering rather than relying on a global threshold.
*   **S2:** As a result of utilizing conformal prediction, the method is model-agnostic and requires no access to the model’s internal logits.
*   **S3:** The experiments are diverse and demonstrate improved performance across multiple domains.
* **S4:** The paper is clearly written.

**Weaknesses:**

*   **W1:** The paper lacks an analysis of sensitivity to the choice of hyperparameters, such as $\alpha$ and $\rho$. It would be helpful to discuss their effects and empirically demonstrate it.
*   **W2:** The authors did not report the loss of the remaining (not filtered) synthetic samples, which could empirically illustrate that the proposed approach controls the risk.
*   **W3:** The method’s effectiveness heavily depends on the quality of the surrogate scores, but this aspect is not discussed (or perhaps I missed it).

**Questions:**

1. Could the authors justify the use of the margin $|p(y∣x)−0.5|$ as a filtering score in the image task, given that it does not provide a meaningful ranking of samples?
2. Given the formulation in Section 3.3 and the way this section is presented, where the gold-standard score $A$ is unavailable and a learned surrogate $\hat{A}$ is used instead, I would have expected a corollary or lemma characterizing how the risk-control guarantees are affected under this approximation. Could the authors provide a theoretical result for this case (which I believe is quite straightforward), or is this extension intended as a heuristic only?

---

> ### Author Response · Authors · 2025-11-24
> **Response to Reviewer RqAx**
>
> We thank the reviewer for their time and their appraisal of our paper. We hope the following answers will clear any further doubt or question they have regarding our method.
>
>
> **W1: Lack of evaluation of the method’s sensitivity to the choice of $\lambda$ and $\rho$:**  Thank you for pointing this out --- this is a very valid criticism of our initial draft. We have added experiments to this effect (see Table 3 of the supplementary material) to  illustrate how the choice of hyperparameters affects the diversity of the filtered data. As expected, the sensitivity varies with the dataset. Intuitively, a larger value of $\lambda$ and a smaller value of $\rho$ will lead to lower diversity --- an effect that is confirmed in Table 3: in the Diagnosis dataset for instance, the Shannon Entropy of the samples shows a clear decreasing trend with $\lambda$ (e.g. when $\rho=1$, from 8.14 for $\lambda =0.3$  to  7.81 for $\lambda =0.9$  ), and increasing with $\rho$ (for $\lambda=0.5$, the Shannon entropy increases from 7.42 for $rho=0$ to 8,13 for $\rho=2$). These hyperparameters should thus be selected to balance diversity with faithfulness to the original data distribution.
>
>
>
>
> &nbsp;
>
>
> **W2: Loss on the remaining samples**  Thank you for the suggestion. We have now included empirical violation rates in the supplementary section of the updated manuscript (Figure 7), which directly illustrate the risk control achieved by our method.
>
> &nbsp;
>
> **W3: Dependency of the method on the surrogate scores:** The effectiveness of the method does indeed depend on the quality of the surrogate scores. When the surrogate is of lower quality, the procedure becomes more conservative, as it will filter more samples and yield less power. The resulting guarantees remain valid due to the distribution-free nature of conformal prediction --- we view this as an added benefit of our method.
>
>
> &nbsp;
>
>
> **Q1: Choice of the margin:** For the image task, the margin |p(y|x) - 0.5| is built to provide low scores to images for which the classifier is unsure, and higher scores for images for which the classifier is more sure. This score seemed reasonable to us, insofar as the classification accuracy is fairly low, even without additional samples (>75%).
>
> &nbsp;
>
> **Q2: Impact of the Surrogate: **  The key to understanding this proxy is via an analogy with standard conformal prediction: one may calibrate using $|E[Y|X] - \hat{f}(X)|$ or $|Y - \hat{f}(X) | = |E[Y|X] + noise - \hat{f}(X) |$, where the second uses a noisy but unbiased proxy for E[Y|X] and yields a conservative but valid interval. Our setting is analogous: when the surrogate is a noisy measurement of the gold-standard score, conformal calibration on the surrogate produces a conservative upper bound on the true risk. This provides the needed performance guarantee for Section 3.3.

---

> > ### Comment · Reviewer_RqAx · 2025-11-26
> >
> > Thank you for your response, specifically for including the empirical risk figure and the sensitivity analysis with respect to $\lambda$ and $\rho$. In Figure 7, where the authors present the risk, it seems that the risk is conservatively controlled; I suggest that the authors comment on why this occurs in these experiments in the text.
> >
> > Regarding the dependence of the method on surrogate scores, this is indeed a benefit when utilizing conformal prediction, due to its model-agnostic nature and the freedom to choose the score function. For a conformal prediction audience, this is clear; however, I noted that it might be worth explicitly mentioning, as the audience is not necessarily familiar with this.
> >
> > My comments and questions were appropriately addressed, and I maintain my positive score.

---

### Note · Program_Chairs · 2026-01-17
**Submission Desk Rejected by Program Chairs**

The following references in this submission do not refer to real documents and/or have major errors in bibliographic information:

 Sai Rajeswar, Tianyi Lin, Ruihan Bao, Andrew Saxe, and R. Devon Hjelm. Diversity and fidelity in generative models: Trade-offs and metrics. arXiv preprint arXiv:2301.06712, 2023.